# Deep Sprite-based Image Models: An Analysis

**Zeynep Sonat Baltacı**  *sonat.baltaci@enpc.fr*
*LIGM, CNRS, Univ Gustave Eiffel, ENPC, Institut Polytechnique de Paris, France*

**Romain Loiseau**  *romain.loiseau@enpc.fr*
*LIGM, CNRS, Univ Gustave Eiffel, ENPC, Institut Polytechnique de Paris, France*

**Mathieu Aubry**  *mathieu.aubry@enpc.fr*
*LIGM, CNRS, Univ Gustave Eiffel, ENPC, Institut Polytechnique de Paris, France*

**Reviewed on OpenReview:** *https: // openreview. net/ forum? id= pXuxMLFo9g*

## Abstract

While foundation models drive steady progress in image segmentation and diffusion algorithms compose always more realistic images, the seemingly simple problem of identifying recurrent patterns in a collection of images remains very much open. In this paper, we focus on sprite-based image decomposition models, which have shown some promise for clustering and image decomposition and are appealing because of their high interpretability. These models come in different flavors, need to be tailored to specific datasets, and struggle to scale to images with many objects. We dive into the details of their design, identify their core components, and perform an extensive analysis on clustering benchmarks. We leverage this analysis to propose a deep sprite-based image decomposition method that performs on par with state-of-the-art unsupervised class-aware image segmentation methods on the standard CLEVR benchmark, scales linearly with the number of objects, identifies explicitly object categories, and fully models images in an easily interpretable way.[1]

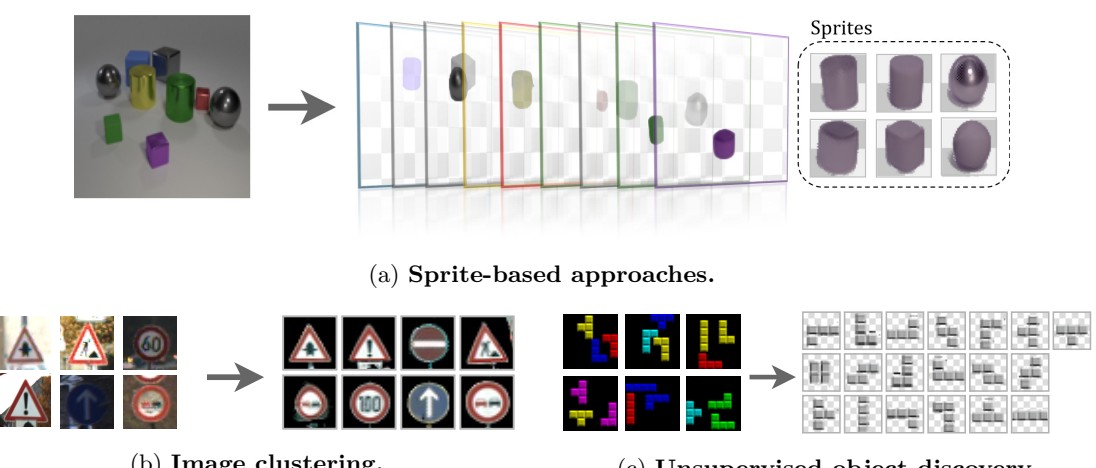

(a) **Sprite-based approaches.**

(b) **Image clustering.**

(c) **Unsupervised object discovery.**

Figure 1: (a) Sprite-based approaches take a set of images as input and learn jointly a family of sprites and how to decompose each image into a sequence of transformed sprites. They can be applied to (b) image clustering and (c) unsupervised object discovery.

---

[1] https://github.com/sonatbaltaci/deepsprite

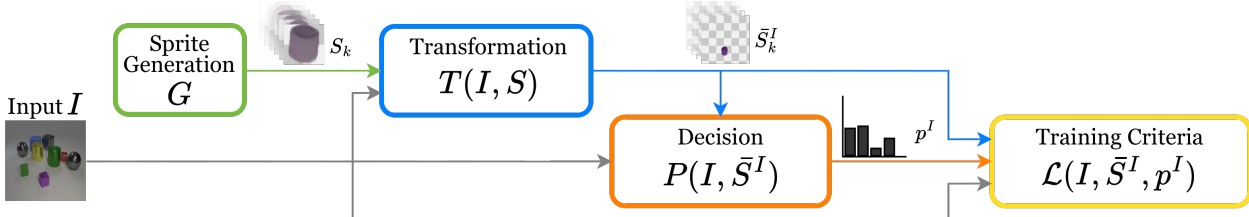

Figure 2: **Overview.** We decompose all sprite-based models in four main components: (1) a *Sprite Generation Module* (●) that outputs $K$ sprites $S$, (2) a *Transformation Module* (●) that takes as input an image $I$ and the sprites $S$ to predict transformed sprites $\bar{S}^I$, (3) a *Decision Module* (●) that takes the image $I$ and transformed sprites $\bar{S}^I$ as input and outputs a probability distribution $p^I$ for using the sprites, and (4) a *Training Criteria* (●) which consist of a reconstruction loss and potential regularization terms.

# 1 Introduction

Identifying recurring patterns in an image collection is a task in which humans excel. It is also critical for many scientific applications, from historical documents to medical image analysis. Although foundation features or models might be attractive tools for approaching this problem, they come with their black-box effects and the biases of their training data. Instead, we advocate for methods that can be directly optimized on the target image collection, offer maximal interpretability, and have limited bias.

In this study, we focus more specifically on sprite-based methods (Visser et al., 2019; Monnier et al., 2020; 2021; Smirnov et al., 2021; Loiseau et al., 2024; Siglidis et al., 2024), which are the main type of object-centric approaches to unsupervised object discovery that allow joint categorization and localization (Villa-Vásquez & Pedersoli, 2024) (Fig. 1). Sprite-based methods offer several other attractive advantages. First, they explicitly model repeated patterns as a finite set of prototypical objects, called sprites. Second, not only do they provide for each analyzed image a layered decomposition, but they also give direct, explicit access to the transformation of the sprites in the image, such as position, scale, and color transformations. Third, their relationship with the standard K-means clustering algorithm (MacQueen, 1967; Bottou & Bengio, 1994) and transformation invariant methods (Frey & Jojic, 1999; 2001; 2003) is well understood (Monnier et al., 2020). However, sprite-based methods have not been fully explored. In particular, the impact of architectural changes and training methodology on their results is poorly understood and different approaches have been demonstrated on different non-standard datasets. Our goal in this study is to better identify key design choices for sprite-based methods and analyze their effects.

In more detail, we separate sprite-based architectures into their key components, visualized in Fig. 2: Sprite Generation Module, Transformation Module, Decision Module, and Training Criteria. For each components, we identify different design choices proposed in the literature, as well as simpler baselines, detailed in Fig. 3. We explain how the training criteria correspond to different image composition models and are related to the exponential cost of some sprite-based image decomposition approaches. We show that one can effectively study the impact of most design choices for clustering, where the benchmarks are more realistic and diverse than for image decomposition, where they are mainly synthetic.

Our key insight is that the main challenge of sprite-based approaches lies in jointly learning and selecting the sprites. K-means-style optimization for sprite selection leads to the discovery of more visually coherent, precise, and semantically accurate objects, without the need for complex regularization, as regularization is implicitly enforced through cluster reassignment policies. However, this type of optimization scales exponentially with the number of objects per image. We show that different regularization techniques can improve approaches that directly predict sprite selection. While we perform most of our analyses on the more diverse and less computationally demanding clustering benchmarks, this actually enable us to design an approach which we demonstrate can generalize to multi-layer decomposition.

This paper is organized as follows: First, in Section 2, we review the literature on clustering and image decomposition. Second, in Section 3, we present a unified formalization for sprite-based image decomposition

models. Third, in Section 4, we perform a comparative analysis of the different design choices on clustering and propose our new approach. Finally, in Section 5, we extend and evaluate our approach for multi-layer image decomposition.

**Contributions.** Our contributions are as follows:

- We perform an exhaustive analysis of sprite-based methods and identify their key components.
- We systematically study their impact on clustering benchmarks.
- We propose a novel sprite-based approach that predicts sprite selection and scales linearly with the number of objects per image.

## 2 Related Work

### 2.1 Image Clustering

We focus on image clustering approaches that are most related to our work and classify them into pixel-based and deep-feature-based clustering. For a broader literature review, we refer the reader to dedicated surveys (Zhou et al., 2024; Ren et al., 2024; Wei et al., 2024).

#### 2.1.1 Pixel-based Clustering

Clustering in pixel space is highly challenging since the image content can be associated with different backgrounds and can undergo spatial and color transformations that completely change its pixel representation. Traditional clustering methods, such as K-means (MacQueen, 1967), therefore lead to limited results when applied directly on full images. EM-based transformation-invariant clustering algorithms have been proposed to gain invariance to user-defined families of transformation (Frey & Jojic, 1999; 2001; 2003). They operate directly in image space, compare pixel values, and provide prototypical representations of clusters. The idea of transformation invariance was also adopted in congealing-based image alignment models that learn transformations using a data-driven approach (Cox et al., 2008; 2009; Huang et al., 2007; Miller et al., 2000; Annunziata et al., 2019; Learned-Miller, 2006), some with a focus on clustering (Mattar et al., 2012; Liu et al., 2009). Deep Transformation-invariant (DTI) Clustering builds on this idea but optimizes prototypes and transformations in a deep learning framework Monnier et al. (2020). The sprite-based image models we study are very related to DTI-Clustering, which can be seen as a single-layer image model, where sprites correspond to prototypes.

#### 2.1.2 Deep Features and Clustering

Many recent deep architectures adopt clustering as an objective for representation learning, e.g., Caron et al. (2018; 2020); Li et al. (2021); Liang et al. (2023), without specifically targeting clustering performance. More relevant to us are those that specifically target clustering, focusing on various technical tools, such as CNNs (Yang et al., 2016; Chang et al., 2017), autoencoders (Xie et al., 2016; Mrabah et al., 2019; Dizaji et al., 2017; Kosiorek et al., 2019; Shaham et al., 2018), mutual information (Hu et al., 2017; Ji et al., 2019), generative models (Jiang et al., 2016; Mukherjee et al., 2018), or instance discrimination (Niu et al., 2022; Van Gansbeke et al., 2020). The crucial and common aspect of these deep clustering approaches is relying on abstract image features. However, relying on deep representations of images and clusters in feature space makes it very hard to interpret the results, performance, and failures, especially in a visually intuitive way.

### 2.2 Image Decomposition

Image decomposition is a broad concept and could encompass broad areas of research from image co-segmentation to layered video representations. In this section, we focus on the approaches that are the most relevant to our work and are often referred to as unsupervised multi-object segmentation approaches or deep object-centric image decomposition methods. We only review single-image methods, and do not dive into the many works that leverage motion, video, or 3D. We follow the taxonomy of Karazija et al. (2021),

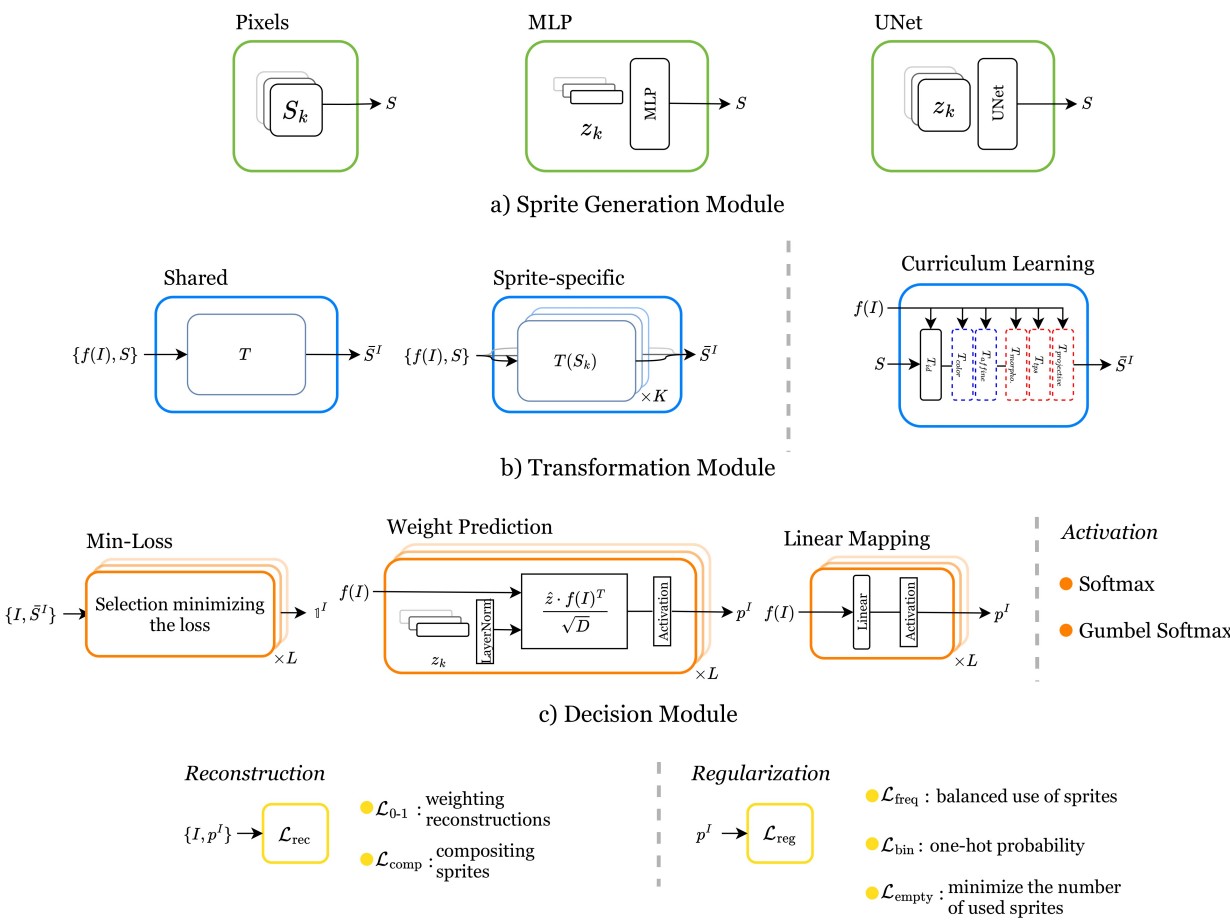

a) Sprite Generation Module

b) Transformation Module

c) Decision Module

d) Composition Model and Training Criteria

Figure 3: **Possible design choices** for the main components identified in Fig. 2. Modules can take as input the input image $I$, features from the input image $f(I)$, the sprites $S$, the transformed sprites $\bar{S}^I$ and the predicted sprite probabilities $p^I$. (a) The *Sprite Generation Module* (●) can learn the sprites directly as learnable parameters (*Pixels*), generate them from learnable latent variables with a multi-layer perceptron (*MLP*) or a *UNet* architecture. (b) The *Transformation Module* (●) parameters can be learned with a *shared* or *sprite-specific* network, and with different *curriculum learning* strategies. (c) The *Decision Module* (●) can select sprites leading to the minimum reconstruction error (***Min-Loss***), or predict them using the sprites' latent representations (***Weight Prediction***), or directly a linear projection (***Linear Mapping***), with alternative activations. (d) The *Composition Model and Training Criteria* (●), where the main loss can either be the sum of the reconstruction errors obtained with all the possible sprites selection weighted by their probability ($\mathcal{L}_{0-1}$) or the reconstruction error with composite sprites ($\mathcal{L}_{\text{comp}}$). It can also include regularizations ($\mathcal{L}_{\{\text{freq, bin, empty}\}}$).

differentiating pixel-based, glimpse-based, and sprite-based approaches. Another view of these approaches is presented in Greff et al. (2020), which differentiates approaches depending on the type of slot they rely on, namely instance slots, sequential slots, spatial slots, and category slots. For a broader review of unsupervised object discovery approaches, we refer the reader to Villa-Vásquez & Pedersoli (2024).

**Pixel-based methods.** Pixel-based methods assign each pixel to an image component, typically by performing probabilistic pixel clustering. Early works tackle this clustering problem by developing approaches based on denoising autoencoders (DAE) (Greff et al., 2015; Vincent et al., 2008), Iterative Amortized Group-

ing (Greff et al., 2016), and Neural Expectation Maximization (Greff et al., 2017). However, these pioneer methods were limited to simple images with a small number of objects.

More recent models following this pixel-based paradigm include MONet (Burgess et al., 2019), IODINE (Greff et al., 2019), eMORL (Emami et al., 2021), and GENESIS (Engelcke et al., 2020; 2021), and demonstrate results on the more challenging synthetic CLEVR dataset of rendered 3D spheres, cubes, and cylinders. They typically output a segmentation mask for each image component as well as a latent code that enables generating an appearance image for each component. The ECON (von Kügelgen et al., 2020) method is built on MONet but is more related to our work because it explicitly models object layers and is designed to completely model occluded objects. However, it has only been demonstrated on very simple synthetic data.

Moving away from probabilistic scene representation and pixel clustering of these so-called scene-mixture models, Locatello et al. (2020) proposed a discriminative approach to scene component identification. Their slot attention mechanism localizes scene components through an iterative clustering-like attention mechanism and leads to a latent representation for each slot, which encodes both its mask and appearance. This approach has been successfully applied to perform object discovery on much more challenging images. DINOSAUR (Seitzer et al., 2023) first demonstrated results on real-world datasets by applying slot attention to DINO features (Caron et al., 2021), instead of pixels. It has also been combined with more complex slot decoders, including auto-regressive (Singh et al., 2022; Kakogeorgiou et al., 2024) and diffusion (Jiang et al., 2023; Wu et al., 2023; Singh et al., 2025) ones.

**Glimpse-based methods.** Glimpse-based methods first extract regions of the image containing objects and then predict object models for each region. This idea was introduced by the Attend, Infer, Repeat approach (AIR) (Eslami et al., 2016), which was the main inspiration for a series of works, such as SQAIR (Kosiorek et al., 2018), SPAIR (Crawford & Pineau, 2019), or SuPAIR (Stelzner et al., 2019). Similar to early pixel-based approaches, these works were developed for very simple synthetic datasets. More recent works, such as SPACE (Lin et al., 2020), GNM (Jiang & Ahn, 2020), and AST (Sauvalle & de La Fortelle, 2023), extend glimpse-based approaches to CLEVR-like datasets. However, they seem to be out-shone by slot-attention-based approaches, which brought comparable efficiency to pixel-based approaches.

**Sprite-based methods.** Sprite-based methods learn a set of object prototypes, referred to as sprites, and how to combine sprites to reconstruct images. These sprites make their image model much more tangible than pixel and glimpse-based approaches, enabling them to discover object categories, not just instance segmentation. StampNet (Visser et al., 2019) can be considered as the first deep sprite-based approach. It learns a latent space to categorize and localize objects, but was only demonstrated on very simple synthetic datasets. Capsule approaches (Kosiorek et al., 2019; Xiang et al., 2021) are similar in spirit and have been categorized as sprite-based methods in Villa-Vásquez & Pedersoli (2024). However, they learn abstract feature-based representations of object parts, they are typically evaluated on clustering benchmarks, and to the best of our knowledge, they have not been demonstrated on standard multi-object datasets. Appealing results on video-game and text images have been demonstrated by MarioNette (Smirnov et al., 2021), which sees sprite discovery as a self-supervised learning problem, and learns to predict sprite occurrence and position. DTI-Sprites (Monnier et al., 2021) models sprite shape and color transformation, enabling it to tackle more complex datasets, including CLEVR data. However, it requires testing many sprite configurations instead of predicting sprite occurrence, and thus does not scale to a large number of objects. Focusing on text line analysis, the Learnable Typewriter (Siglidis et al., 2024) combines ideas from MarioNette and DTI-Sprites for applications in digital humanities. Similar ideas have been applied to model 3D point clouds (Loiseau et al., 2024). Our analysis encompasses all these sprite-based approaches and clarifies their differences.

## 3 Sprite-based Approaches

In this section, we first present a unified view of sprite-based decomposition methods for clustering and layered image decomposition, summarized in Figure 2. Then, for each key component that we identify, we

Table 1: **Comparison of sprite-based models.** Existing sprite-based methods make very different choices for several of the four components that we identified, making direct comparison between their performance difficult. Our exhaustive analysis leads to an informed choice of all the components for clustering (Ours-C) and image decomposition (Ours-D).

| Method | ● Sprite Generation | ● Curriculum | ● Sharing | ● Decision | ● Activation | ● $\mathcal{L}_{\text{rec}}$ | ● $\mathcal{L}_{\text{reg}}$ | ⧉ |
|---|---|---|---|---|---|---|---|---|
| StampNet Visser et al. (2019) | pixels | all | sprite-specific | linear | hard GS | $\mathcal{L}_{\text{comp}}$ | - | decomposition |
| DTI-Clustering Monnier et al. (2020) | pixels | one-by-one | sprite-specific | Min-Loss | none | $\mathcal{L}_{\text{comp=0-1}}$ | reassignment | clustering |
| DTI-Sprites Monnier et al. (2021) | pixels | one-by-one | sprite-specific | Min-Loss | softmax | $\mathcal{L}_{\text{comp=0-1}}$ | reassignment, $\mathcal{L}_{\text{empty}}$ | decomposition |
| MarioNette Smirnov et al. (2021) | MLP | all | shared | weight prediction | none | $\mathcal{L}_{\text{comp}}$ | $\mathcal{L}_{\text{bin}}$ | decomposition |
| Learnable Earth Parser Loiseau et al. (2024) | 3D point clouds | one-by-one | shared | linear | softmax | $\mathcal{L}_{0-1}$ | $\mathcal{L}_{\text{freq}}$ | decomposition |
| Learnable Typewriter Siglidis et al. (2024) | MLP | all | shared | weight prediction | softmax | $\mathcal{L}_{\text{comp}}$ | - | decomposition |
| **Ours-C** | MLP | one-by-one | sprite-specific | linear | soft GS | $\mathcal{L}_{\text{comp}}$ | $\mathcal{L}_{\{\text{freq,bin}\}}$ | clustering |
| **Ours-D** | MLP | one-by-one | sprite-specific | linear | soft GS | $\mathcal{L}_{\text{comp}}$ | $\mathcal{L}_{\text{empty}}$ | decomposition |

detail different design choices that have been introduced in the literature and that we consider in our study, which are summarized in Figure 3, and are related to the literature in Table 1.

### 3.1 Unified View and Formalization

Our key insight is that sprite-based approaches rely on four main components that we present first. We then discuss how these modules can be used for multi-object image decomposition and clustering. The choices made by different sprite-based approaches in the literature are summarized in Table 1.

#### 3.1.1 Key Components

Sprite-based approaches take as input an image $I \in \mathbb{R}^{W \times H \times C}$, with $C = 1$ for a grayscale image and $C = 3$ for an RGB image, and predict a set of layers associated with this image. As visualized in Figure 2, we identified four key components in sprite-based methods:

- A **sprite generation module**, $G$ (Section 3.2), which learns $K$ sprites $S_1, \cdots, S_K$, with for all $k \in \{1, \cdots, K\}$, $S_k \in \mathbb{R}^{R \times R \times C'}$, where $R$ is the size of the sprites and $C'$ is the number of channels per sprite. Sprites can be interpreted as prototypical images and can include segmentation, encoded as a transparency mask. Depending on the approach, $C'$ can be 1 (a grayscale image), 2 (a grayscale image and transparency), 3 (an RGB image), or 4 (an RGB image with a transparency channel).

- A **transformation module**, $T$ (Section 3.3), which takes as input a target image $I$ and sprites $S_1, \cdots, S_K$, and outputs a set of transformed sprites $\bar{S}^I$. Transformation typically includes color and spatial transformations. The transformed sprites are images of the same size as the input image $I$, with an optional transparency channel. Note that this module can predict several transformations for each sprite, enabling the modeling of images with multiple elements, as we clarify in Section 3.1.2.

- A **decision module**, $P$ (Section 3.4), which predicts probabilities $p^I$ for each of the transformed sprites to be used in the reconstruction of the input image.

- A **reconstruction loss**, $\mathcal{L}$ (Section 3.5), which evaluates how well the transformed sprites associated with the predicted probabilities explain the input image, and with which the model is optimized.

These components and the losses correspond to an image formation model, $\mathcal{C}(\bar{S}, p)$.

#### 3.1.2 Layered Image Decomposition

For layered image decomposition, one typically assumes a maximum number of layers $L$. Each sprite $S_k$ for $k \in \{1, \cdots, K\}$ is then transformed into $L$ sprites $\bar{S}^I_{k,l} \in \mathbb{R}^{W \times H \times C'}$ for $l \in \{1, \cdots, L\}$, with $C' = C + 1$, leading to a set of $K \times L$ transformed sprites $\bar{S}^I = (\bar{S}^I_{1,1}, \cdots, \bar{S}^I_{K,L})$. Sprites are selected according to $p^I \in [0, 1]^{K \times L}$. Note that one of the sprites can be used as an empty sprite, *i.e.*, frozen and completely transparent, to allow modeling a variable number of objects. Background can be modeled using one or several specific opaque sprites, possibly with particular constraints (e.g., having a uniform color) and be

associated with their own specific transformations. To simplify notation, we do not differentiate background sprites from the other sprites. In our experiments on layered image decomposition, we always model the background with a single sprite. The image formation model, $\mathcal{C}$, composites the transformed background sprite with the sprites from the following layers. To better handle occlusion, we follow DTI-Sprites (Monnier et al., 2021) and predict a matrix defining the order of the layers.

### 3.1.3 Clustering

In the case of clustering, the simplest scenario (Monnier et al., 2020) is to consider a single-layer image model using only completely opaque sprites. In that case, the set of transformed sprites is $\bar{S}^I = (\bar{S}^I_{1,1}, \cdots, \bar{S}^I_{K,1})$, with for all $k \in \{1, \cdots, K\}$, $\bar{S}^I_{k,1} \in \mathbb{R}^{W \times H \times C}$ the transformed version of sprite $S_k$. Note that if there are no transformations, and the $L_2$ loss between the input and the transformed sprite that best approximates it is optimized, this model boils down to standard K-means (MacQueen, 1967; Bottou & Bengio, 1994).

Another approach that typically leads to better results for more complex images (Monnier et al., 2021) is to explicitly model the background using a background sprite and the different clusters with sprites including a transparency channel, and thus consider a 2-layer model. The image formation model, $\mathcal{C}$, composites the transformed background sprite with the other transformed sprites depending on the output $p^I \in [0,1]^K$ of the selection module.

Both of these approaches can be seen as specific cases of layered image decomposition and leverage the same modules, enabling us to start our analysis by focusing on the simpler clustering scenario.

### 3.2 ● Sprite Generation Module

The sprites $S_1, \cdots, S_K$ are the visual representation of the recurrent patterns identified by the model in the target image collection. They are thus common to all input images $I$, they are themselves modeled as images – color or grayscale, and associated or not with a transparency mask – and they can be learned with different strategies.

#### 3.2.1 Learning Pixel Values

Directly learning the sprite, *i.e.* setting each sprite's pixel values as learnable parameters, is the simplest choice and has been used in Monnier et al. (2020; 2021).

#### 3.2.2 Decoding Learned Latent Variables with a Generator Network (MLP or U-Net)

Motivated by the possibility of using latent variables to link sprite generation and clustering, Smirnov et al. (2021) proposes to learn $K$ latent vectors $z_1, \cdots, z_K$ and a generation network $G$ that takes as input those latent vectors, and outputs the corresponding sprite $S_k = G(z_k)$. Note that while generated by a network, the sprites still do not depend on the input image $I$, and that once the network is trained, they could be computed once and for all, without using the generator network. Following Siglidis et al. (2024), we explore the use of a Multi-Layer Perceptron (MLP) or a U-Net architecture (Ronneberger et al., 2015) (U-Net) as the generation network.

### 3.3 ● Transformation Module

Sprite-based approaches account for variations in the appearance of objects in terms of shape or color by explicitly modeling them. Given an input image $I$, they predict one (for clustering) or several (for image decomposition) transformations for each sprite. The family of transformations that are available and the way in which they are learned are important hyperparameters, and the optimal choice depends on the target dataset. Transformations typically include (i) spatial transformations, modeled with Spatial Transformer Networks (Jaderberg et al., 2015), and (ii) affine color transformation, where parameters are predicted and applied on the sprite values. They may include more specific transformations, such as morphological transformations to model stroke width for the MNIST dataset (LeCun et al., 2010). There are several key design choices in this transformation learning that we explore.

### 3.3.1 Curriculum Learning

Because transformations could model dramatic changes, curriculum learning is the key to progressively learning meaningful transformations. We explore various curriculum scheduling strategies. To study them, we first decided on a fixed order of transformation by increasing complexity, as visualized in Figure 3b: no transformation, affine color transformation, affine spatial transformation, morphological transformation, Thin Plate Spline (TPS) transformation, and projective transformation. With all transformations initialized as the identity function, we then tested different strategies:

- ***all***: optimizing all transformations together from the start,
- ***id+rest***: learning first without any transformation, then optimizing all transformations together,
- ***id+g1+g2***: grouping transformations into three groups – (id) no transformation, (g1) affine color and spatial transformations, (g2) other transformations – and adding each group of transformations into the optimization one-by-one, and
- ***one-by-one***: adding each of them into the optimization one-by-one.

Note that for each dataset, we only use transformations relevant to the dataset (see Appendix Table 12).

### 3.3.2 Sprite-Specific vs. Shared Transformations

Another question we explore is the possibility and consequences of sharing the transformations among sprites. Intuitively, one could expect the sprites to be better aligned if the same transformations are applied to all sprites, while an architecture that applies specific transformations to all sprites might be more powerful. Sharing transformations might also be beneficial when modeling a large number of sprites.

### 3.4  ● Decision Module

A crucial problem of sprite-based approaches is deciding which transformed sprites to use to reconstruct a specific image. We consider two types of solutions.

### 3.4.1 Minimum Loss

A simple approach is to choose the sprites that minimize the loss (Bottou & Bengio, 1994; Monnier et al., 2020). However, this means that (i) during training, only sprites that are selected receive gradients, and thus some might never be used, which requires specific re-assignment strategies, and (ii) when modeling images with multiple objects, the number of possible sprite combinations is exponential in the number of objects, which complicates optimization. Note that this approach can be seen as a deterministic layer predicting one-hot probability vectors $p^I$, and we refer to it as ***Min-Loss***.

### 3.4.2 Probability Prediction

Another approach is to use a neural network to predict which transformed sprites should be used for a specific input image $I$ by predicting probability distributions among transformed sprites. While this is much more in line with common deep learning paradigms, we show experimentally that jointly learning the sprites, their transformations, and the selection of the best sprites is a challenging optimization problem, which requires using many regularization functions that make the method more specific and less robust.

The more standard architecture to predict such a probability distribution is a network that takes as input the target image $I$ and finishes with a linear layer and a softmax, which we refer to as ***linear mapping***. However, MarioNette (Smirnov et al., 2021) proposes having a network instead predict classification weights from latent variables, shared with the sprite generation module, which are then compared with the input image features, before applying a softmax. We refer to this approach as ***weight prediction***.

Finally, because what is ultimately needed is a binary selection of the sprites, we experimented with replacing the softmax by Gumbel softmax (Jang et al., 2017; Maddison et al., 2017), similar to StampNet (Visser et al., 2019). However, while StampNet uses Gumbel softmax with binary selection, we use Gumbel softmax with soft selection, which consistently led to better performances.

### 3.5 🟡 Composition Model and Training Criteria

We decompose the training loss as a reconstruction loss, $\mathcal{L}_{\text{rec}}$, and a regularization loss, $\mathcal{L}_{\text{reg}}$:

$$\mathcal{L} = \mathcal{L}_{\text{rec}} + \mathcal{L}_{\text{reg}} . \tag{1}$$

We study two reconstruction losses, which actually correspond to two different composition models $\mathcal{C}(\bar{S}, p)$, as well as different regularization losses.

#### 3.5.1 Composition Model and Reconstruction Loss

The composition model, $\mathcal{C}(\bar{S}^I, p^I)$, composites transformed sprites into an image. The first way to see this model is to consider that it can only select sprites in a binary way, and thus, the loss should be a weighted sum of reconstruction errors of each sprite selection weighted by their probability ($\mathcal{L}_{\text{0-1}}$). The second way to build composite sprites is by weighting transformed sprites according to predicted probabilities $p$, then reconstructing images with composite sprites ($\mathcal{L}_{\text{comp}}$).

More formally, let us define $\mathcal{C}^L$ the standard alpha-blending composition of $L$ images $(A_1, \alpha_1), \cdots, (A_L, \alpha_L)$, where for $l = 1, \cdots, L$ the $A_l$ are RGB images and $\alpha_l$ their associates transparency channels:

$$\mathcal{C}^L\left((A_1, \alpha_1), \cdots, (A_L, \alpha_L)\right) = \sum_{l=1}^{L} \left( \alpha_l \prod_{k=l+1}^{L} (1 - \alpha_k) \right) A_l, \tag{2}$$

where the product is 1 if empty and multiplications are to be understood pixelwise. Let us consider probabilities $p^I \in \mathbb{R}^{K \times L}$ and transformed sprites $\bar{S}^I_{k,l} \in \mathbb{R}^{W \times H \times C}$ for all $k \in \{1, \cdots, K\}$ and $l \in \{1, \cdots, L\}$. Then $\mathcal{L}_{\text{0-1}}$ is defined by:

$$\mathcal{L}_{\text{0-1}}(\bar{S}^I, p^I) = \sum_{(k_1, \cdots k_L) \in \{0, \cdots, K\}^L} \left( \prod_{l=1}^{L} p^I_{k_l, l} \right) ||I - \mathcal{C}^L(\bar{S}^I_{k_1, 1}, \cdots, \bar{S}^I_{k_L, L})||_2^2 , \tag{3}$$

and $\mathcal{L}_{\text{comp}}$ is defined by:

$$\mathcal{L}_{\text{comp}}(\bar{S}^I, p^I) = ||I - \mathcal{C}^L(\sum_{k=1}^{K} p^I_{k,1} \bar{S}^I_{k,1}, \cdots, \sum_{k=1}^{K} p^I_{k,L} \bar{S}^I_{k,L})||_2^2 . \tag{4}$$

As can be seen from the equations, $\mathcal{L}_{\text{0-1}}$ requires to compute $K^L$ composite images, which is computationally prohibitive for large numbers of sprites and layers, while $\mathcal{L}_{\text{comp}}$ only requires computing $L$ composed sprites and a single composite images. However, $\mathcal{L}_{\text{comp}}$ corresponds to an image composition model where different transformed sprites can be merged, which might lead to undesired optima where objects are represented by overlapping multiple sprites. Note that in the case where $p^I$ is binary, we have $\mathcal{L}_{\text{0-1}} = \mathcal{L}_{\text{comp}}$.

#### 3.5.2 Regularizations

We consider three regularization losses.

First, $\mathcal{L}_{\text{freq}}$ attempts to prevent some sprites from never being used, by penalizing using a sprite with a frequency lower than a scalar value $\epsilon \in [0, 1]$:

$$\mathcal{L}_{\text{freq}} = \sum_{k=1}^{K} \max\left( 0, \epsilon - \frac{1}{|D|} \sum_{I} \sum_{l=1}^{L} p^I_{k,l} \right) , \tag{5}$$

where in practice the loss is computed over a batch of images $I$. Note that DTI-Sprites (Monnier et al., 2021) has a re-assignment strategy for unused sprites that plays a similar role and has a similar minimum frequency hyperparameter $\epsilon$.

Second, $\mathcal{L}_{\mathrm{bin}}$ encourages one-hot probability vectors $p^I$, and thus attempts to avoid several transformed sprites being used together to reconstruct an object. Thus, it is particularly meaningful to regularize $\mathcal{L}_{\mathrm{comp}}$. Following Smirnov et al. (2021), we define $\mathcal{L}_{\mathrm{bin}}$ by:

$$\mathcal{L}_{\mathrm{bin}} = \frac{1}{K}\sum_{k=1}^{K}\mathrm{Beta}(2,2)\left(p_k^I\right) \ , \tag{6}$$

where the probability density function of the Beta distribution is given by:

$$f(x;\alpha,\beta) = \begin{cases} \frac{x^{\alpha-1}(1-x)^{\beta-1}}{B(\alpha,\beta)} & \text{for } 0 < x < 1 \\ 0 & \text{otherwise} \end{cases}$$

where $\alpha > 0$ and $\beta > 0$ are the shape parameters, and $B(\alpha,\beta)$ is the Beta function, defined as:

$$B(\alpha,\beta) = \int_0^1 t^{\alpha-1}(1-t)^{\beta-1}\,dt = \frac{\Gamma(\alpha)\Gamma(\beta)}{\Gamma(\alpha+\beta)},$$

where, $\Gamma(\cdot)$ is the Gamma function, which generalizes the factorial function $(n-1)!$.

Third, following Monnier et al. (2021), $\mathcal{L}_{\mathrm{empty}}$ encourages the model to use as few sprites as possible to reconstruct an image, and attempts to avoid failure cases like sprites used with a high transparency to better reconstruct details of the images. It penalizes the use of non-empty sprites, and writing $e$ the index of the empty sprite can be defined as:

$$\mathcal{L}_{\mathrm{empty}} = \sum_{l=1}^{L}(1-p_{e,l}^I) \ . \tag{7}$$

$\mathcal{L}_{\mathrm{reg}}$ is defined as a weighted sum of these three regularization losses:

$$\mathcal{L}_{\mathrm{reg}} = \lambda_{\mathrm{freq}}\mathcal{L}_{\mathrm{freq}} + \lambda_{\mathrm{bin}}\mathcal{L}_{\mathrm{bin}} + \lambda_{\mathrm{empty}}\mathcal{L}_{\mathrm{empty}} \ , \tag{8}$$

with $\lambda_{\mathrm{freq}}$, $\lambda_{\mathrm{bin}}$ and $\lambda_{\mathrm{empty}}$ scalar hyperparameters.

## 4 Analysis on Clustering

In this section, we analyze single-layer sprite-based approaches on clustering, for which experiments are faster, and datasets are more diverse than for unsupervised image decomposition, and we leverage this analysis to define a new approach for sprite-based clustering. Section 4.1 introduces the details of our experimental setup. Sections 4.2 to 4.4 present comparative analysis of approaches through experiments on *Sprite Generation*, *Transformation*, *Decision*, and *Training Criteria* (see Fig. 3 for an overview and terminology). Finally, Section 4.5 compares our clustering approach with the state-of-the-art.

### 4.1 Experimental Setup

#### 4.1.1 Datasets

We conducted experiments on 8 datasets with different characteristics: MNIST (LeCun et al., 2010), ColoredMNIST (Arjovsky et al., 2019), FashionMNIST (Xiao et al., 2017), AffNIST (Tieleman, 2013), USPS (Hull, 1994), FRGC (Phillips et al., 2005), SVHN (Netzer et al., 2011), and GTSRB-8 (Stallkamp et al., 2012) (detailed in the Appendix .1). Digit datasets (LeCun et al., 2010; Arjovsky et al., 2019; Hull, 1994; Netzer et al., 2011) differ in complexity, ranging from grayscale digits to real-world RGB street number images. The other datasets tackle fashion items (Xiao et al., 2017), faces (Phillips et al., 2005), and traffic signs (Stallkamp et al., 2012), offering a diversity of challenges.

Table 2: ● **Analysis of the sprite generation module for clustering.** We report the performances of sprite generation approaches. We report accuracy (%) and standard error over 10 runs. ●: one-by-one, sprite-specific transformation, ●: Min-Loss, ●: $\mathcal{L}_{\text{comp}=0-1}$, reassignment.

| Module | Init. | MNIST | ColoredMNIST | FashionMNIST | AffNIST | USPS | FRGC | SVHN | GTSRB-8 | Average |
|--------|-------|-------|--------------|--------------|---------|------|------|------|---------|---------|
| *Pixel Space* | | | | | | | | | | |
| Pixels | sample | **97.2±0.0** | 94.5±1.5 | 58.3±0.6 | 93.3±2.0 | 86.3±2.0 | 40.4±0.8 | 42.8±2.4 | **51.4±1.5** | 70.5 |
| Pixels | random | **97.2±0.0** | **95.5±1.3** | 57.2±0.7 | 89.5±2.0 | 84.0±0.5 | **40.8±0.7** | 42.2±2.0 | 51.2±0.6 | 69.7 |
| *Latent Space* | | | | | | | | | | |
| MLP | random | 97.1±0.0 | 94.3±1.5 | **58.9±0.7** | **95.7±1.3** | 85.5±2.3 | 40.3±0.4 | **45.8±1.2** | 51.1±1.6 | 71.1 |
| UNet | | 97.1±0.1 | 94.8±1.5 | 57.9±1.3 | 94.5±1.8 | **86.6±1.5** | 33.7±1.8 | 45.8±2.4 | 50.3±1.0 | 70.1 |

### 4.1.2 Training Details and Evaluation

We report the mean accuracy over all samples for clustering using Hungarian matching (Kuhn, 1955) for cluster-to-class assignments. To evaluate the impact of our regularization losses, we provide a sensitivity analysis in Appendix .2.2 over a sequential grid search for the regularization weights. This analysis shows that performance variance across the searched ranges is low, establishing that while $\mathcal{L}_{\text{freq}}$ is crucial to prevent cluster collapse, the model remains robust to minor hyperparameter variations. Details of the training setup and complete hyperparameter configurations are provided in Appendix .2. Unless stated otherwise, we report the mean and standard error over 10 runs for each experiment.

### 4.1.3 Reference Setting

We sequentially evaluate the influence of each of the key component we have identified, starting from the DTI-Clustering setting (Monnier et al., 2020), which demonstrated competitive results for clustering, and which is closest to the K-means baseline. Note that our notion of a sprite encompasses the notion of prototype used in DTI-Clustering. Then, in each section, we define a new reference setting for each of our component, depending on our experimental analysis.

### 4.2 ● Sprite Generation Module

As detailed in Section 3.2 and Figure 3a, we compare directly learning pixel values and learning sprites through a generator network. When learning pixel values, we compare initializing the sprites randomly or from a random sample, similar to the original DTI-Clustering (Monnier et al., 2020). When learning sprites in latent space, we compare using a two-layer MLP and a UNet. For the MLP, we learn a latent representation of size 128 and use a hidden layer with 128 units. For UNet, we used a latent representation with the same dimension as the sample sprite and the architecture of Siglidis et al. (2024).

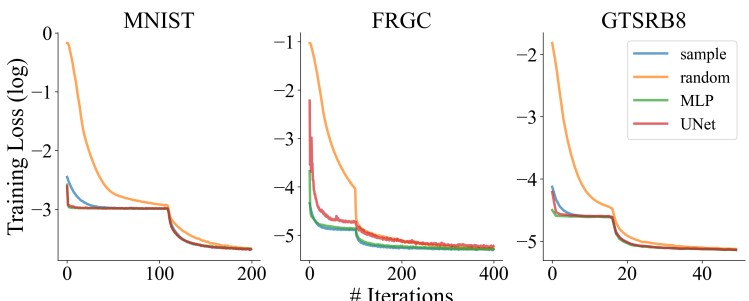

Figure 4: ● **Training loss for different sprite generation modules.** We show the average loss over 10 runs for 3 datasets. For all datasets, learning sprites through a generator network converges faster. Better seen in the digital version.

Our results, reported in Table 2, show that the best performing approach depends on the dataset. Learning sprites with an MLP leads to slightly better results on average, it is within or close to the error margin of the best method on all datasets, and clearly improving on pixel-based approaches on AffNIST and SVHN. Moreover, an analysis of the training loss curves shown in Fig. 4 shows that learning sprites through generator networks leads to clearly faster convergence. We thus adopt **learning sprites through an MLP for the rest of our analysis.**

### 4.3 ● Transformation Module

As explained in Section 3.3, we explore different constraints on the deformation module, namely different curriculum and weight-learning strategies.

#### 4.3.1 Curriculum Learning

Table 3: ● **Effect of curriculum learning on the transformation module.** We explore different curriculum strategies. We report accuracy (%) and standard deviation over 10 runs. ●: MLP, ●: sprite-specific transformation, ●: Min-Loss, ●: $\mathcal{L}_{comp=0-1}$, reassignment.

| | *Curriculum strategy* | | | |
|---|---|---|---|---|
| Dataset | all | id+rest | id+g1+g2 | one-by-one |
| MNIST | 88.1±2.6 | 95.8±1.1 | 95.8±0.9 | **97.1±0.0** |
| ColoredMNIST | 82.8±2.7 | 83.9±2.4 | 94.2±1.8 | **94.3±1.5** |
| FashionMNIST | 56.0±1.2 | 58.7±1.8 | 57.8±0.9 | **58.9±0.7** |
| AffNIST | 83.1±4.0 | 81.7±3.0 | **95.8±1.4** | 95.7±1.3 |
| USPS | 81.3±2.1 | **86.0±1.8** | 83.2±1.2 | 85.5±2.3 |
| FRGC | 34.9±0.9 | 34.5±0.5 | 39.1±0.5 | **40.3±0.4** |
| SVHN | 32.6±2.2 | 33.3±0.4 | 45.8±1.2 | 45.8±1.2 |
| GTSRB-8 | 49.3±1.5 | **51.3±1.8** | 51.1±1.6 | 51.1±1.6 |
| Average | 63.5 | 65.7 | 70.4 | 71.1 |

In Table 3, we report results using various curriculum strategies to learn the transformations presented in Section 3.3. They show that curriculum is critical for good performance. A 2-step-only curriculum, which can be interpreted as a K-means initialization followed by a full unfreeze of the network, is not sufficient, while splitting transformations into two groups already leads to good results. One-by-one curriculum performs best, and we thus continue using **one-by-one curriculum for the rest of our analysis.**

#### 4.3.2 Shared Transformations

Table 4: ● **Effect of sharing transformations among sprites in the transformation module.** We report accuracy (%) and standard deviation over 10 runs. ●: MLP, ●: one-by-one, ●: Min-Loss, ●: $\mathcal{L}_{comp=0-1}$, reassignment.

| Dataset | Shared transfo. | Sprite-specific transfo. |
|---|---|---|
| MNIST | 91.9±2.2 | **97.1±0.0** |
| ColoredMNIST | 92.6±2.0 | **94.3±1.5** |
| FashionMNIST | 57.0±0.4 | **58.9±0.7** |
| AffNIST | 86.4±2.8 | **95.7±1.3** |
| USPS | 84.4±2.3 | **85.5±2.3** |
| FRGC | **41.1±0.6** | 40.3±0.4 |
| SVHN | 34.3±0.1 | **45.8±1.2** |
| GTSRB-8 | 49.2±1.2 | **51.1±1.6** |
| Average | 67.1 | 71.1 |

Sharing transformations among sprites would intuitively put them in the same "reference frame" which would be beneficial for qualitative analysis. We visualize this effect in Fig. 5 on the ColoredMNIST and AffNIST datasets. When transformations are not shared, sprites have non-uniformed colors and positions, while they are much more consistent when transformations are shared. Although this qualitative property would be desirable, we found in the quantitative results reported in Table 4 that sharing transformations significant deteriorates quantitative performances. **We thus keep sprite-specific transformations for each sprite in the rest of the analysis.**

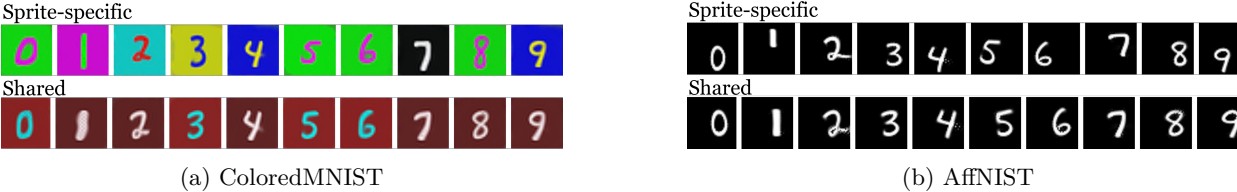

Figure 5: ● **Qualitative effect of sharing transformations among sprites in the transformation module.** We compare on (a) ColoredMNIST (Arjovsky et al., 2019) and (b) AffNIST (Tieleman, 2013) the sprites learned with sprite-specific transformations (top rows) with the ones learned with shared transformations (bottom rows). Sharing the transformations among sprites encourages them to be more uniform, e.g., have similar (a) colors and (b) spatial location.

## 4.4 ● Decision Module and ● Training Criteria

Training criteria and decision modules are closely related. Thus, we first analyze jointly decision module and training criteria, then study the impact of regularizations.

### 4.4.1 Reconstruction Loss and Decision

Table 5: **Results of different decision modules (●) and training criteria (●).** We experimented with the training criteria and decision modules, along with Gumbel softmax. We report accuracy (%) and standard deviation over 10 runs. We train all models and the baseline (second row) until convergence, which might mean a different number of iterations for different models. ●: MLP, ●: one-by-one, sprite-specific transformation.

| $\mathcal{L}_{\text{rec}}$ | $p_k$ | MNIST | ColoredMNIST | FashionMNIST | AffNIST | USPS | FRGC | SVHN | GTSRB-8 | Average |
|---|---|---|---|---|---|---|---|---|---|---|
| $\mathcal{L}_{\text{comp}} = \mathcal{L}_{0-1}$ | Min-Loss | 92.4±1.6 | 71.0±2.1 | 58.3±0.6 | 89.2±2.2 | 82.1±1.7 | 30.2±0.7 | **47.1±1.8** | 47.1±1.1 | 64.7 |
| | *w/ reassignment* | 96.5±0.5 | **92.0±2.0** | 59.6±0.7 | **97.3±0.0** | **88.4±2.9** | **41.1±0.6** | 43.3±2.7 | 49.3±1.3 | 70.9 |
| $\mathcal{L}_{0-1}$ | weight prediction | 86.6±1.2 | 42.0±3.2 | 55.3±0.9 | 66.6±4.1 | 79.9±1.4 | 11.2±0.6 | 33.1±0.9 | **51.7±0.2** | 53.3 |
| | linear mapping | 88.8±1.5 | 33.0±4.9 | 54.7±1.1 | 55.5±1.7 | 73.7±3.1 | 17.9±0.7 | 31.1±1.3 | 51.6±0.4 | 50.8 |
| $\mathcal{L}_{\text{comp}}$ | weight prediction | 72.4±0.9 | 50.5±3.6 | 35.1±1.3 | 72.7±2.2 | 54.3±2.1 | 40.2±0.9 | 19.7±0.3 | 38.2±1.0 | 47.9 |
| | *w/ Gumbel softmax* | 93.2±1.6 | 47.7±4.5 | 60.6±0.4 | 75.8±1.4 | 82.1±0.2 | 38.7±0.8 | 34.7±0.7 | 50.3±0.1 | 60.4 |
| | linear mapping | 72.1±1.4 | 47.5±1.6 | 36.3±1.1 | 66.9±0.9 | 54.3±2.0 | 40.4±1.1 | 19.9±0.5 | 38.5±0.1 | 47.0 |
| | *w/ Gumbel softmax* | **96.5±0.1** | 53.2±4.3 | **60.7±0.8** | 75.4±2.3 | 82.2±0.4 | 39.5±1.3 | 33.9±0.5 | 50.0±0.2 | 61.4 |

In Table 5, we compare the reconstruction losses we introduced – namely $\mathcal{L}_{0-1}$ defined in Eq. (3) and $\mathcal{L}_{\text{comp}}$ defined in Eq. (4) – alongside the different decision modules. *Min-Loss* selection, for which both losses are the same, using a cluster re-assignment strategy (Monnier et al., 2020) (Table 5 row 2) shows overall better performance than training the network to predict the sprite selection. This higher performance is largely due to the implicit regularization given by the empty cluster reassignment strategy proposed in (Monnier et al., 2020) (Table 5 rows 1 and 2).

Qualitatively, the main failure case of $\mathcal{L}_{\text{comp}}$ is to compose a layer from several sprites, as can be seen in Fig. 6a for MNIST, where a 9 digit is reconstructed using a circle and a loop, and in Fig. 6b for FRGC, where different sprites are combined to model lighting effects. As optimizing reconstruction by composition is not the targeted behavior for clustering, we experimented with replacing softmax activation with Gumbel softmax for this $\mathcal{L}_{\text{comp}}$, both with linear mapping and weight prediction (Table 5 rows 6 and 8). This resulted in a significant improvement in performances of more than 10% on average. While performances remains lower than with *Min-Loss* selection with reassignment by almost 10%, this led to the best results with a predicted sprite selection, almost on par with *Min-Loss* selection without re-assignment regularization. Learning sprite selection is appealing as it does not require to test all selection possibilities, as in *Min-Loss* selection, which will be prohibitively costly when using multiple layers. Because we obtained slightly better performances with linear mapping than classification weight prediction, and because it is conceptually

$\mathcal{L}_{0-1}$

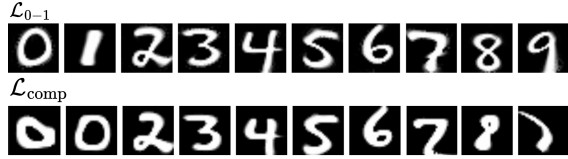

$\mathcal{L}_{\mathrm{comp}}$

$\mathcal{L}_{0-1}$

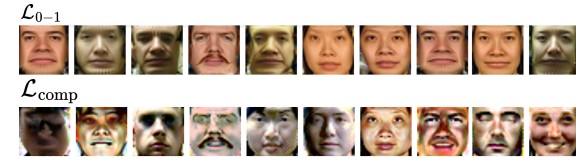

$\mathcal{L}_{\mathrm{comp}}$

(a) MNIST LeCun et al. (2010)        (b) FRGC Phillips et al. (2005)

Figure 6: ● **Qualitative results with different training criteria.** Compared with weighting the reconstruction loss for each sprite ($\mathcal{L}_{0-1}$, top rows), weighting transformed sprites and composing to reconstruct ($\mathcal{L}_{\mathrm{comp}}$, bottom rows) results in (a) sprites representing parts of the objects instead of the object itself and (b) sprites focusing on the distinct characteristics of a subject and using composition to model shading effects.

Table 6: ● **Effect of regularization.** Experiments on regularization losses with two training criteria and Gumbel softmax. We report accuracy (%) and standard deviation over 10 runs. ●: MLP, ●: one-by-one, sprite-specific transformation, ●: linear mapping.

| $\mathcal{L}_{\mathrm{rec}}$ | $\mathcal{L}_{\mathrm{freq}}$ | $\mathcal{L}_{\mathrm{bin}}$ | MNIST | ColoredMNIST | FashionMNIST | AffNIST | USPS | FRGC | SVHN | GTSRB-8 | Average |
|---|---|---|---|---|---|---|---|---|---|---|---|
| $\mathcal{L}_{0-1}$ | - | - | 88.8±1.5 | 33.0±4.9 | 54.7±1.1 | 55.5±1.7 | 73.7±3.1 | 17.9±0.7 | 31.1±1.3 | 51.6±0.4 | 50.8 |
| | ✓ | - | **98.2±0.0** | 93.1±2.1 | 57.6±1.5 | **97.1±0.1** | 82.8±0.1 | 41.1±0.6 | **38.0±2.0** | **57.5±0.2** | 70.7 |
| | *w/ softmax* | | | | | | | | | | |
| | - | - | 72.1±1.4 | 47.5±1.6 | 36.3±1.1 | 66.9±0.9 | 54.3±2.0 | 40.4±1.1 | 19.9±0.5 | 38.5±0.1 | 47.0 |
| | ✓ | - | 78.5±1.6 | 48.4±3.1 | 40.6±1.1 | 75.5±0.0 | 54.3±2.0 | 42.6±1.1 | 23.4±0.9 | 38.8±0.3 | 50.3 |
| | ✓ | ✓ | 95.3±0.5 | 81.7±3.5 | **62.0±1.5** | 83.1±0.0 | 63.1±2.2 | 42.6±1.1 | 34.4±0.5 | 54.5±2.1 | 64.6 |
| $\mathcal{L}_{\mathrm{comp}}$ | *w/ Gumbel softmax* | | | | | | | | | | |
| | - | - | 96.5±0.1 | 53.2±4.3 | 60.7±0.8 | 75.4±2.3 | 82.2±0.4 | 39.5±1.3 | 33.9±0.5 | 50.0±0.2 | 61.4 |
| | ✓ | - | 96.7±0.0 | 95.9±0.1 | 60.7±0.8 | 94.1±1.9 | 82.2±0.4 | **44.8±0.8** | 35.3±0.4 | 53.2±1.2 | 70.4 |
| | ✓ | ✓ | 96.7±0.0 | **96.0±0.1** | 60.7±0.8 | 94.1±1.9 | **85.3±1.1** | **44.8±0.8** | 37.6±0.3 | 53.2±1.2 | 71.1 |

simpler, **we use linear mapping for the rest of the paper**, and explore if its performance could be further improved using additional regularization losses.

### 4.4.2 Regularization Losses

We report in Table 6 the results obtained with using $\mathcal{L}_{\mathrm{freq}}$ (Eq. (5)) and $\mathcal{L}_{\mathrm{bin}}$. Note that $\mathcal{L}_{\mathrm{bin}}$ is designed to overcome the composition issue associated to $\mathcal{L}_{\mathrm{comp}}$, we do not test it with $\mathcal{L}_{0-1}$, and $\mathcal{L}_{\mathrm{empty}}$ does not make sense for clustering, where no empty sprite is used. We selected the regularization loss weights through a grid search for each dataset to optimize performance.

Using $\mathcal{L}_{\mathrm{freq}}$ (Eq. (5)) as a regularization significantly increases performance. Both when using $\mathcal{L}_{0-1}$ and $\mathcal{L}_{\mathrm{comp}}$ losses coupled with Gumbel softmax, this leads to results on par with *Min-Loss* selection with reassignment (Table 6). This again validates our claim that the superior performance of *Min-Loss* selection is largely due to the implicit regularization of the reassignment strategy.

To improve results obtained with $\mathcal{L}_{\mathrm{comp}}$ we evaluated using $\mathcal{L}_{\mathrm{bin}}$ (Eq. (6)) to encourage binary selection, similar to Smirnov et al. (2021). $\mathcal{L}_{\mathrm{bin}}$ significantly improves the results with normal softmax while remaining worse than the best approaches, and provides a small improvement when using Gumbel softmax which already encourages binary selection. **We thus propose using $\mathcal{L}_{\mathbf{comp}}$ with Gumbel softmax, and $\mathcal{L}_{\mathbf{freq}}$ and $\mathcal{L}_{\mathbf{bin}}$ regularizations.**

### 4.5 Comparison with State-of-the-art

Given the analysis of the previous sections, we use the following design choices, summarized in Table 1, for clustering: using an MLP-based sprite generation module, with sprite-specific transformations, learned one-by-one in a curriculum fashion, a linear decision module with Gumbel softmax, a composite reconstruction loss, and frequency and binning regularization. We compare the performance of this setting with a single opaque layer (Ours-C 1 layer) to a variety of competing clustering methods in Table 7. For SVHN and

Table 7: **Comparisons on clustering.** We compare our results with methods that cluster over features as well as pixels. We report accuracy (%) and standard deviation for our method over 10 runs.

| Method | # runs | MNIST | ColoredMNIST | FashionMNIST | AffNIST | USPS | FRGC | SVHN | GTSRB-8 |
|---|---|---|---|---|---|---|---|---|---|
| *Clustering on learned features* | | | | | | | | | |
| JULE Yang et al. (2016) | 3 | 96.4 | - | 56.3 | - | 95.0 | 46.1 | - | - |
| DEPICT Dizaji et al. (2017) | 5 | 96.5 | - | 39.2 | - | **96.4** | **47.0** | - | - |
| DSCDAN Yang et al. (2019) | 10 | 97.8 | - | **66.2** | - | 86.9 | - | - | - |
| *+ with data augmentation and/or ad-hoc data representation* | | | | | | | | | |
| SpectralNet Shaham et al. (2018) | 5 | 97.1 | - | - | - | - | - | - | - |
| IMSAT Hu et al. (2017) | 12 (5) | 98.4 | (10.6) | - | (18.2) | - | - | **57.3** | 26.9 |
| ADC Haeusser et al. (2019) | 20 | **98.7** | - | - | - | - | 43.7 | 38.6 | - |
| SCAE Kosiorek et al. (2019) | 5 | **98.7** | - | - | - | - | - | 55.3 | - |
| IIC Ji et al. (2019) | 5 | 98.4 | 10.6 | - | **57.6** | - | - | - | - |
| SCAN Van Gansbeke et al. (2020) | 5 | - | - | - | - | - | - | 54.2 | **90.4** |
| *Clustering on pixels* | | | | | | | | | |
| K-means | 10 | 54.8 | - | 54.1 | - | 65.3 | 22.7 | 12.2 | - |
| DTI-Clustering Monnier et al. (2020) | 10 | **97.3** | **96.8** | **61.2** | **95.5** | **86.4** | 39.6 | 44.5 | - |
| **Ours-C** 1 layer | 10 | 96.7±0.0 | 96.0±0.1 | 60.7±0.8 | 94.1±1.9 | 85.3±1.1 | **44.8±0.8** | 37.6±0.3 | 53.2±1.2 |
| *+ multi-layer* | | | | | | | | | |
| DTI-Sprites Monnier et al. (2021) | 10 | - | - | - | - | - | - | **63.1** | **89.9** |
| **Ours-C** 2 layers | 10 | - | - | - | - | - | - | 52.4±0.5 | 80.9±1.8 |

GTSRB-8, we also report our approach using a background model (Ours-C 2 layers). Note that most approaches rely on learning and clustering features, which limit the results' interpretability, and that many leverage specific data-augmentation or representations, such as Gabor filters, which are strong priors and simplify the task. Our results are competitive with state-of-the-art on class-aware metrics, while predicting cluster selection, and learning an explicit cluster prototype and image-specific transformation. While it often performs slightly worse than DTI-Clustering, our setting does not rely on comparing each possible reconstruction to the target to assign clusters, but instead directly learns and predicts cluster selection. Thus, as shown in the next session, our approach can be directly extended into an efficient multi-layer image decomposition model.

# 5 Analysis for Multi-layer Image Decomposition

In this section, we explore how our analysis of sprite-based image models for clustering can be leveraged for multi-layer image decomposition. In the following, we first summarize our experimental setting, including datasets and metrics, and then discuss our results.

## 5.1 Experimental Setup

### 5.1.1 Datasets

We present results on the Tetrominoes (Greff et al., 2019) – images of 3 non-overlapping colored 2D blocks sampled among 19 unique ones, on black background –, the Multi-dSprites (Kabra et al., 2019) – images of up to 5 possibly overlapping colored 2D objects of different sizes sampled among 3 unique ones, on gray background –, and the CLEVR6 and CLEVR (Johnson et al., 2017) datasets – images of respectively up to 6 and 10 possibly overlapping colored 3D objects of different sizes sampled among 6 unique ones, on a simple background. More details are given in the Appendix .1. Note that all of these datasets are synthetic and relatively simple, but they are the main ones used in the literature for our task.

### 5.1.2 Training Details

Details of the transformations for the foreground and background are given in the Appendix Table 12. Our empirical results across four distinct datasets show that the regularization weight $\lambda_{\text{empty}}$ is crucial for multi-layer object decomposition, and higher scene complexity generally demands higher $\lambda_{\text{empty}}$. Details of training setup and hyperparameters for each dataset are provided in the Appendix .2.

Table 8: **Results for multi-object semantic discovery.** (●) Sprite generation, (●) decision and activation function, (●) training criterion and regularization. Mean accuracy (mAcc) and average mean IoU (avg-mIoU) over classes, with standard error over 3 runs. (†): longer training, except Monnier et al. (2021) on CLEVR (in *italic*) obtained with the training schedule in the paper. Gray entries denote single-run results where initial performance indicated lower performance than the baseline.

| Method | ● | ● | ● | ● | ● | Tetrominoes | | Multi-dSprites | | CLEVR6 | | CLEVR | |
|---|---|---|---|---|---|---|---|---|---|---|---|---|---|
| | | | | | | mAcc | avg-mIoU | mAcc | avg-mIoU | mAcc | avg-mIoU | mAcc | avg-mIoU |
| DTI-Sprites Monnier et al. (2021)† | Pixels | Min-Loss | S | $\mathcal{L}_{0-1=comp}$ | $\mathcal{L}_{empty}$ | **99.5±0.2** | **99.2±0.3** | **91.3±0.9** | **84.0±1.4** | **79.3±2.7** | **64.2±3.1** | *69.8±5.0* | ***55.7±4.3*** |
| **Ours-D** | MLP | linear mapping | GS | $\mathcal{L}_{comp}$ | - | 74.3±1.4 | 64.4±1.9 | 65.6±0.1 | 54.4±0.1 | 66.8±0.4 | 49.6±0.8 | 59.3±7.4 | 43.0±5.6 |
| | | | | | $\tau$ | 92.7±3.9 | 89.2±5.5 | 65.2 | 53.8 | 56.3 | 39.6 | 55.3 | 42.1 |
| | | | | | $+ \mathcal{L}_{freq}$ | 93.9±0.4 | 89.9±0.5 | - | - | - | - | - | - |
| | | | | | $\mathcal{L}_{empty}$ | - | - | 65.9±0.5 | 54.7±0.4 | 74.7±1.4 | 57.8±1.5 | **70.6±0.1** | 55.3±0.1 |
| | | | | | $+ \mathcal{L}_{freq}$ | - | - | 66.0±1.0 | 54.7±0.9 | 72.2±1.2 | 54.8±1.3 | 70.5±1.2 | 53.9±0.5 |
| | | | | | $+ \mathcal{L}_{bin}$ | - | - | 65.4 | 54.2 | 65.0 | 46.5 | 68.0 | 53.1 |

### 5.1.3 Metrics

For our analysis in Table 8, we reported two class-aware metrics, mean accuracy (mAcc) and average mean Intersection-over-Union (avg-mIoU). We use Hungarian matching to align the predicted and ground-truth classes. Mean accuracy measures the proportion of correctly-predicted pixels in classification. The average mean IoU computes the IoU, a segmentation accuracy metric, class-wise and averages it across all classes, including background, to reflect class awareness.

To give results comparable with the metrics most frequently reported in the literature, we also report in Table 9 instance mean IoU (mIoU) and the Adjusted Rand Index computed only for the foreground (ARI-FG). Instance mean IoU measures the segmentation accuracy of predicted instances without considering the class of the prediction, but takes the background into account. ARI-FG evaluates how well pixels' instance assignments align with the ground truth while ignoring the background. These two metrics are adopted by the literature because most existing methods focus solely on predicting instance segmentation without providing the corresponding class labels. For the few approaches that additionally provide class predictions, we also report mean accuracy (mAcc) and mean IoU averaged over classes (avg-mIoU).

## 5.2 Results

### 5.2.1 Regularizations

In Table 8, we analyze the impact of different regularizations on the performance of our approach and compare it to DTI-Sprites (Monnier et al., 2021). Indeed, the regularization needs are different from the ones in clustering. In particular, $\mathcal{L}_{bin}$, which we adopted for clustering, prevents multiple sprites from the same layer from being combined to reconstruct different parts of the same object, but it does not prevent the same effect with sprites from different layers. Thus, in addition to $\mathcal{L}_{bin}$ and $\mathcal{L}_{freq}$, we experimented with $\mathcal{L}_{empty}$ (Eq. (7)) which favors the use of an empty sprite, *i.e.* a sprite with a completely transparent alpha mask. Because we observed early-stage high-confidence class predictions during training, which is likely detrimental to learning, we also explored the impact of learning the Gumbel softmax temperature parameter, $\tau$, which could mitigate this effect.

For Tetrominoes, where the number of objects is constant and equal to the number of layers, $\mathcal{L}_{empty}$ and $\mathcal{L}_{bin}$ make little sense, and we only explore learning the Gumbel softmax temperature $\tau$ and $\mathcal{L}_{freq}$, while we explore all regularizations for the other datasets.

Learning the Gumbel softmax temperature $\tau$ gives a huge boost to the results on Tetrominoes. One of the three runs actually matches the almost perfect performance of DTI-Sprites, emphasizing the additional complexity of learning the class prediction. Adding $\mathcal{L}_{freq}$ to learning $\tau$ further improves the average on Tetrominoes, but they remain below the almost perfect results of DTI-Sprites, without any run matching it. In contrast, for Multi-dSprites, CLEVR6, and CLEVR, learning $\tau$ leads to the worst results.

For Multi-dSprites, CLEVR6, and CLEVR, we thus discarded learning $\tau$ and instead experimented with $\mathcal{L}_{\text{empty}}$, trying to better estimate the number of objects, which is the main challenge for our approach without regularization on CLEVR6 and CLEVR. On Multi-dSprites (Greff et al., 2019), which includes 3 distinct objects (square, ellipsoid, and heart), our method is still clearly outperformed by DTI-Sprites (Monnier et al., 2021), due to the fact that our model fails to discover a distinct representation of the heart shape, instead reconstructing it as a composition of two rotated ellipsoids. On CLEVR6 and CLEVR, our performance with $\mathcal{L}_{\text{empty}}$ is on par with DTI-Sprites, and the results are qualitatively similar. Further adding $\mathcal{L}_{\text{freq}}$ does not significantly change these results, and adding $\mathcal{L}_{\text{bin}}$ degrades them. The main difference between our approach and DTI-Sprites on this more challenging CLEVR dataset is the higher scalability of our approach, which we discuss in the next section.

### 5.2.2 Complexity

The major advantage of our model with respect to DTI-Sprites is that it learns to predict which sprite to select for which layer, rather than iteratively trying many sprite selections and combinations, which results in a significant improvement in time complexity, as shown in Fig. 7 on the CLEVR dataset.

The time complexity of our model scales linearly with the maximum number of objects in a scene, while the DTI-Sprites scales exponentially. Note that due to this exponential time complexity, we trained DTI-Sprites in Table 8 and Table 9 with the training schedule in the original paper, *i.e.* 351,900 iterations, while we could train our model until convergence, for 703,800 iterations, in less time.

### 5.2.3 Comparisons

In 9, we compare our results with the state-of-the-art on the CLEVR dataset. AST-Seg-B3-CT (Sauvalle & de La Fortelle, 2023) clearly dominates in terms of mIoU and ARI-FG, but our results are on par with most baselines for these metrics, which focus on instances and do not consider the class prediction. We thus also compared class aware metrics for methods from which can be extracted. For MarioNette (Smirnov et al., 2021), we match learned sprites in the dictionary to classes in a many-to-one manner with Hungarian matching. Because it is the best-performing instance segmentation method, we also applied K-means with $K = 6$ to the object features ($z_{\text{what}}$) of AST-Seg-B3-CT (Sauvalle & de La Fortelle, 2023) and clustered them,

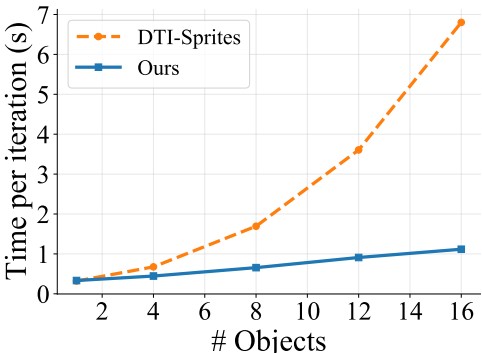

Figure 7: **Complexity.** The time per iteration of our approach scales linearly with the number of object layers, while that of the only other method with comparable results, DTI-Sprites (Monnier et al., 2021), scales exponentially.

leading to a class-aware adaptation of this method. The only method that achieves similar results to ours for class-aware metrics is DTI-Sprites (Monnier et al., 2021), while the other two baselines lag far behind. This emphasizes another advantage of our approach.

### 5.2.4 Qualitative Results

Qualitative examples of our decomposition with a large number of objects from CLEVR (Johnson et al., 2017) are presented in Fig. 8. They show that our model is able to recover both accurate instances and semantic segmentation with a large number of objects.

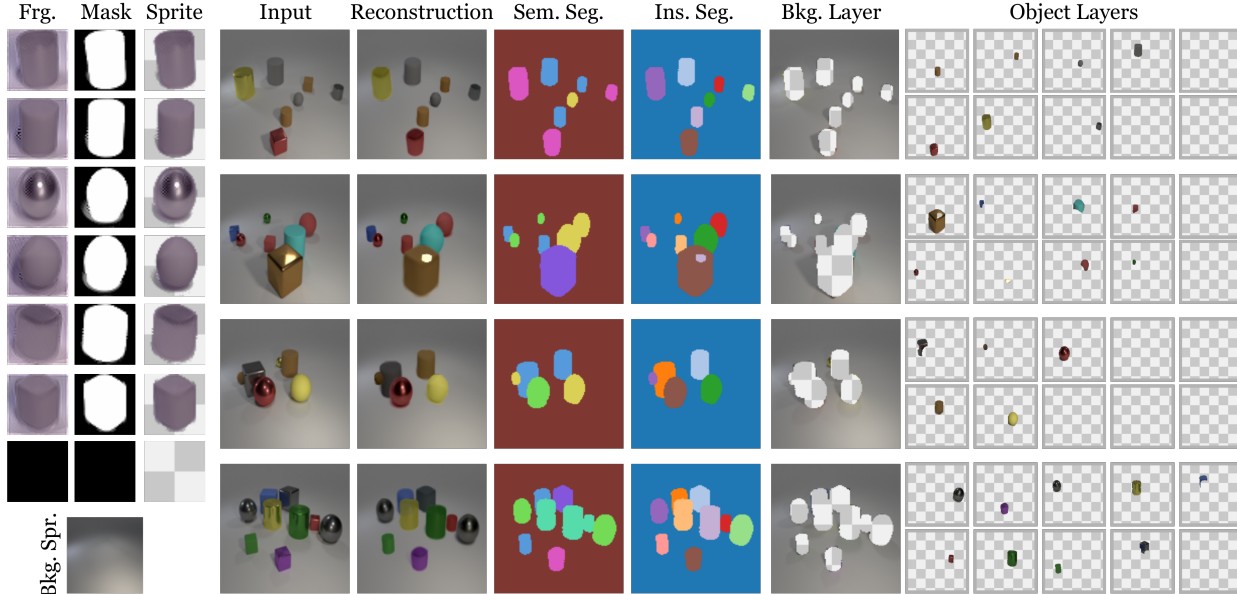

Figure 8: **Qualitative results for multi-object discovery on CLEVR (Johnson et al., 2017).** The three left columns show the sprites' appearances (Frg.), masks, and combination (Sprite), including the empty sprite, and the background. The other columns show for four different examples, the input image, its reconstruction, semantic segmentation (Sem. Seg.), instance segmentation (Ins. Seg.), background (Bkg. Layer), and the different transformed sprites (Object Layers).

Table 9: **Comparisons for instance segmentation** with standard deviation over 3 runs. Sources for † (excluding Monnier et al. (2021)): Karazija et al. (2021) and Sauvalle & de La Fortelle (2023).

| Method | class-aware | CLEVR | | | |
|---|---|---|---|---|---|
| | | mIoU† | ARI-FG† | mAcc | avg-mIoU |
| MONet Burgess et al. (2019) | | 30.7±14.9 | 54.5±11.4 | - | - |
| IODINE Greff et al. (2019) | | 45.1±17.9 | 93.8±0.8 | - | - |
| SPAIR Crawford & Pineau (2019) | | 66.0±4.0 | 77.1±1.9 | - | - |
| GNM Jiang & Ahn (2020) | | 59.9±3.7 | 65.1±4.2 | - | - |
| Slot Attention Locatello et al. (2020) | | 36.6±24.8 | 95.9±2.4 | - | - |
| eMORL Emami et al. (2021) | | 50.2±22.6 | 93.3±3.2 | - | - |
| Genesis-V2 Engelcke et al. (2021) | | 9.5±0.6 | 57.9±20.4 | - | - |
| MarioNette Smirnov et al. (2021) | ✓ | 72.1±0.6 | 56.8±0.4 | 16.1±0.2 | 7.3±0.4 |
| AST-Seg-B3-CT Sauvalle & de La Fortelle (2023) | | 90.3±0.2 | 98.3±0.1 | 20.8±1.2 | 12.1±0.2 |
| DTI-Sprites Monnier et al. (2021) | ✓ | 54.5±1.2 | 93.2±2.0 | 69.8±4.5 | 55.7±6.0 |
| **Ours-D** | ✓ | 53.8±0.3 | 95.1±0.5 | 70.6±0.2 | 55.3±0.2 |

## 6 Conclusion

In this work, we introduced a unified formalization for sprite-based models, specifying their relationships, and unifying approaches for clustering and multi-layer image decomposition. This analysis clarifies the design space of methods in the literature and enables its exploration on the clustering task, which is less computationally intensive and uses more diverse, realistic datasets. This yields in turn an approach to image decomposition that learns to predict sprite selection, to avoid an exponential complexity in the number of objects per image, and maintains strong performance.

Our study offers several key insights for layer-based image decomposition: **(i)** learning sprite representations via an MLP yields slightly better reconstruction, but more importantly significantly accelerates training; **(ii)** sharing transformation parameters across sprites acts as a structural regularizer, encouraging uniformity in color and spatial alignment, but can slightly hurt performances; **(iii)** opposite to the exponential cost of min-

loss optimization, predicting sprite probabilities and using a loss on composite sprites scales linearly with the number of objects per image, enabling to model a larger number of objects per image, but requires explicit regularization and performs slightly worse in simple cases. Moreover, our analysis emphasizes that while the standard CLEVR benchmarks seem saturated in terms of the commonly reported class-agnostic metrics, it is actually far from being the case for class-aware metrics, where our method provides state-of-the-art but imperfect results, encouraging more works in this direction.

One of the main limitations of the approach we propose is the necessity of hyperparameter selection. Although we follow standard protocols in the literature for evaluation and show some robustness to hyperparameter selection, our reliance on ground-truth validation labels to perform a grid search for regularization hyperparameters departs from a fully unsupervised deployment setting where labels are unavailable.

## Acknowledgments

Z. S. Baltacı and M. Aubry supported by the ANR project VHS ANR-21-CE38-0008, and the ERC project DISCOVER funded by the European Union's Horizon Europe Research and Innovation program under grant agreement No. 101076028. This work was granted access to the HPC resources of IDRIS under the allocation AD011015415R1, AD011015415, and AD011014404 made by GENCI. We would like to thank Ioannis Siglidis for insightful discussions, and Robin Champenois and Ségolène Albouy for their contributions to the codebase.

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

### .1 Dataset Descriptions

**MNIST (LeCun et al., 2010)**   MNIST is a widely used dataset of handwritten grayscale digits, containing 60,000 training images and 10,000 testing images.

**ColoredMNIST (Arjovsky et al., 2019)**   Colored MNIST is built from the MNIST dataset by randomly adding color to the foreground and background, resulting in a collection of 70,000 images. Each digit image is transformed into a 3-channel representation, offering a more complex dataset.

**FashionMNIST (Xiao et al., 2017)**   FashionMNIST is designed as an alternative to MNIST, consisting of 60,000 fashion item images for training and 10,000 for testing. These images are grayscale and categorized into 10 classes.

**AffNIST (Tieleman, 2013)**   Derived from MNIST, the AffNIST dataset enriches the original dataset by applying affine transformations to its digits. We employ the test split of 10,000 images to assess algorithm robustness against various transformations.

**USPS (Hull, 1994)**   The United States Postal Service (USPS) dataset includes handwritten grayscale digit images of envelopes, containing 7,291 training samples and 2,007 testing samples.

**FRGC (Phillips et al., 2005)**   The Face Recognition Grand Challenge (FRGC) dataset is a collection of face images in RGB space, which contains over 50,000 images of various individuals captured under different poses, expressions, and lighting conditions.

**SVHN (Netzer et al., 2011)**   The Street View House Numbers (SVHN) dataset includes more than 600,000 RGB images of house numbers captured from Google Street View. It is intended for digit recognition tasks and offers more challenging variations in terms of font styles, sizes, and cluttered backgrounds compared to MNIST.

**GTSRB-8 (Stallkamp et al., 2012)**   The German Traffic Sign Recognition Benchmark (GTSRB) dataset subset (GTSRB-8) focuses on eight common traffic sign classes and contains more than 25,000 images for training and testing.

**Tetrominoes (Greff et al., 2019)**   Tetrominoes contains around 60,000 images with size 35x35 featuring 3 Tetris-like shapes with different color and position from 19 unique shapes. Each image has a black background, and shapes do not occlude each other.

**Multi-dSprites (Kabra et al., 2019)**   Multi-dSprites contains around 60,000 images with multiple oval, heart, or square-shaped objects with a uniform background. Each object has different scale, color, and position, and the maximum number of objects in an image is 5.

**CLEVR (Johnson et al., 2017)**   CLEVR dataset contains 6 unique objects with varying scale, color, and position on a uniform background. Although released for visual reasoning tasks, it is commonly used in object discovery. We reported results in 2 versions of CLEVR: CLEVR6 and CLEVR where the maximum numbers of objects in an image are 6 and 10, respectively. CLEVR6 contains around 35,000 and CLEVR contains around 100,000 images.

### .2 Training Details

We adopt the training setup of Monnier et al. (2020) for clustering and Monnier et al. (2021) for multi-object semantic discovery as our baseline. Hyperparameters are provided in Tables 10 and 11. For Table 8, we report the mean and standard error of 3 runs. Due to its computational complexity, we adopt the training schedule reported for CLEVR6 in Monnier et al. (2021) to CLEVR for DTI-Sprites (*italic* in Table 8). To be comparable with the literature (Karazija et al., 2021), we reported the mean and standard deviation of 3 runs for Table 9. The results for DTI-Sprites and our variation are reported over the whole dataset.

Table 10: **Training setup and hyperparameters for clustering.**

| Dataset | MNIST | ColoredMNIST | FashionMNIST | AffNIST | USPS | FRGC | SVHN | GTSRB-8 |
|---|---|---|---|---|---|---|---|---|
| *Model & Data* | | | | | | | | |
| # sprites | 10 | 10 | 10 | 10 | 10 | 20 | 10 | 8 |
| sprite tr. | id, aff, mor, tps | id, color, aff, tps | id, color, aff, tps | id, aff, mor, tps | id, color, aff, tps | id, color, aff, tps | id, color, proj | id, color, proj |
| sprite tr. curr. | 10, 30, 40 | 10, 30, 60 | 10, 30, 50 | 10, 40, 50 | 120, 240, 400 | 100, 400, 800 | 16, 144 | 160, 1440 |
| *Training* | | | | | | | | |
| batch size | 128 | 128 | 128 | 128 | 128 | 128 | 128 | 128 |
| learning rate | 1e-3 | 1e-3 | 1e-3 | 1e-4 | 1e-3 | 1e-3 | 1e-3 | 1e-3 |
| weight decay | 1e-6 | 1e-6 | 1e-6 | 1e-6 | 1e-6 | 1e-6 | 1e-6 | 1e-6 |
| lr. step | 70 | 90 | 70 | 74 | 500 | 1300 | 240 | 2400 |
| # epochs | 80 | 100 | 80 | 90 | 640 | 1400 | 264 | 2640 |
| $\lambda_{\text{freq}}$ | 0.01 | 0.1 | 0 | 0.01 | 0 | 0.01 | 0.01 | 0.1 |
| $\lambda_{\text{bin}}$ | 0 | 0.001 | 0 | 0 | 0.01 | 0 | 0.001 | 0 |

Table 11: **Training setup and hyperparameters for multi-object decomposition.**

| Dataset | Tetrominoes | Multi-dSprites | CLEVR6 | CLEVR |
|---|---|---|---|---|
| *Model & Data* | | | | |
| # sprites | 19 | 3 | 6 | 6 |
| # bkg | 0 | 1 | 1 | 1 |
| # objects | 3 | 5 | 6 | 10 |
| # channels | 3 | 3 | 3 | 3 |
| frg., bkg., mask curr. | 600, 0, 1 | 0, 0, 20 | 0, 0, 80 | 0, 0, 80 |
| sprite/layer init. | cons, cons, gauss. | cons, cons, gauss. | cons, mean, gauss. | cons, mean, gauss. |
| init. values | 0.9, 0.9, 0. | 0.9, 0.5, 0. | 0.9, 0., 0. | 0.9, 0., 0. |
| gauss. std. | 5 | 7 | 10 | 10 |
| sprite tr. | id | id, scale+rot. | id, proj. | id, proj. |
| bkg. tr. | - | color | color | color |
| layer tr. | color, scale+affine | color, scale+affine | color, scale+affine | color, scale+affine |
| sprite tr. curr. | - | 40 | 300 | 300 |
| sprite size | 24, 24 | 28, 28 | 40, 40 | 40, 40 |
| image size | 35, 35 | 35, 35 | 128, 128 | 128, 128 |
| occlusion | - | ✓ | ✓ | ✓ |
| *Training* | | | | |
| avg. pool | 1, 1 | 1, 1 | 1, 1 | 1, 1 |
| batch size | 32 | 32 | 32 | 32 |
| learning rate | 1e-4 | 1e-4 | 1e-4 | 1e-4 |
| lr. step | 1000, 1200 | 500, 1000 | 500,800 | 500, 800 |
| # epochs | 1220 | 1020 | 900 | 900 |
| $\lambda_{\text{freq}}$ | 1e-3 | 0 | 0 | 0 |
| $\lambda_{\text{bin}}$ | 1e-4 | 0 | 0 | 0 |
| $\lambda_{\text{empty}}$ | - | 1e-4 | 1e-3 | 1e-2 |

### .2.1 Transformation Module

We follow the transformation setup and order in Table 12 according to Monnier et al. (2020; 2021). Table 12 demonstrates three levels of transformations, applied to the sprites, the background and the layers.

Table 12: ● **Transformation setups of datasets.** Transformations are selected and ordered depending on the characteristics of each dataset. Transformations for background and layers are highlighted.

| Dataset | id. | color | affine | morpho. | tps | proj. | scale+rot. | scale+affine |
|---|---|---|---|---|---|---|---|---|
| MNIST | 1 | | 2 | 3 | 4 | | | |
| ColoredMNIST | 1 | 2 | 3 | | 4 | | | |
| FashionMNIST | 1 | 2 | 3 | | 4 | | | |
| affNIST | 1 | | 2 | 3 | 4 | | | |
| USPS | 1 | 2 | 3 | | 4 | | | |
| FRGC | 1 | 2 | 3 | | 4 | | | |
| SVHN | 1 | 2 | | | | 3 | | |
| GTSRB-8 | 1 | 2 | | | | 3 | | |
| Tetrominoes | 1/1 | 1 | | | | | | 2 |
| Multi-dSprites | 1 | 1/1 | | | | | 2 | 2 |
| CLEVR(6) | 1 | 1/1 | | | | | | 2 |

### .2.2 Analysis on Regularization Hyperparameter Tuning

The weights $\lambda_{\mathrm{freq}}$ and $\lambda_{\mathrm{bin}}$ were searched over the range $\{0, 0.01, 0.1, 1.0\}$ and $\{0, 0.001, 0.01, 0.1\}$ in clustering, respectively. For the multi-layer setup, we conduct a sequential grid search in range $\{0, 0.0001, 0.001, 0.01, 0.1\}$ following the order $\lambda_{\mathrm{empty}}$, $\lambda_{\mathrm{freq}}$, and $\lambda_{\mathrm{bin}}$. We provide the sequential grid search results of $\lambda_{\mathrm{freq}}$ and $\lambda_{\mathrm{bin}}$ on four characteristically distinct datasets, ColoredMNIST Arjovsky et al. (2019), FRGC Phillips et al. (2005), SVHN Netzer et al. (2011), and GTSRB-8 Stallkamp et al. (2012), to demonstrate the sensibility of our model to these hyperparameters in Table 13. Our results indicate that $\lambda_{\mathrm{freq}}$ is a critical hyperparameter for preventing cluster collapse. We observed that $\lambda_{\mathrm{bin}}$ acts as a regularizer that improves the probabilities to be one-hot, but has a lower impact on overall performance compared to $\lambda_{\mathrm{freq}}$. Although tuning regularization hyperparameters via ground truth labels allows us to establish a performance upper bound for the proposed architecture, we acknowledge that this protocol departs from a strictly unsupervised setting. We identify the development of robust, fully unsupervised model selection criteria as a significant remaining challenge for the field.

Table 13: ● **Effect of $\lambda_{\mathbf{freq}}$ (left) and $\lambda_{\mathbf{bin}}$ (right)** on four datasets: ColoredMNIST Arjovsky et al. (2019), FRGC Phillips et al. (2005), SVHN Netzer et al. (2011), and GTSRB-8 Stallkamp et al. (2012). (●) Gumbel softmax, (●) $\lambda_{\mathrm{bin}} = 0$ (left), $\mathcal{L}_{\mathrm{rec}}^{s}$.

| Dataset | $\lambda_{\mathrm{freq}}$ | | | |
|---|---|---|---|---|
| | 1.0 | 0.1 | 0.01 | 0 |
| ColoredMNIST | 82.7±2.2 | **95.9±0.1** | 86.4±2.6 | 53.2±4.3 |
| FRGC | - | 36.1±0.8 | **44.8±0.8** | 39.5±1.3 |
| SVHN | - | 32.8±0.7 | **35.3±0.4** | 33.9±0.5 |
| GTSRB-8 | 53.0±1.15 | **53.2±1.2** | 50.5±0.7 | 50.0±0.2 |

| Dataset | $\lambda_{\mathrm{freq}}$ | $\lambda_{\mathrm{bin}}$ | | |
|---|---|---|---|---|
| | | 0.01 | 0.001 | 0 |
| ColoredMNIST | 0.1 | 56.6±9.3 | **96.0±0.1** | 95.9±0.1 |
| FRGC | 0.01 | 42.5±1.0 | 41.7±0.6 | **44.8±0.8** |
| SVHN | 0.01 | 36.1±0.5 | **37.6±0.3** | 35.3±0.4 |
| GTSRB-8 | 0.1 | 49.7±0.3 | 50.0±0.8 | **53.2±1.2** |

