# OpenReview forum: "Deep Sprite-based Image Models: An Analysis"
_TMLR — Accepted by TMLR_

### Review · Reviewer_8ARL · 2026-02-18

**Summary Of Contributions:**

The paper presents a systematic study of sprite based image models for unsupervised clustering and multi-object image decomposition. It unifies prior work into four components—sprite generation, transformation, decision, and training criteria. Further they conduct extensive ablations across eight clustering datasets to get effective design choices. Building on these insights, they propose a deep sprite-based method that predicts sprite selection, achieves accuracy comparable to reassignment-based Min-Loss selection in clustering, and is extended to multi-layer decomposition with claimed linear scaling in the number of objects.

**Audience:**

Yes

**Audience Explanation:**

The proposed linear-scaling selection mechanism is important for extending sprite models to realistic multi-object scenes.
Potential impact areas where black-box foundation features may be undesirable.

**Broader Impact Concerns:**

There are no ethical implications.

**Claims And Evidence:**

Yes

**Claims Explanation:**

1. The paper offers a clear, unified decomposition of sprite-based methods into four modules, clarifying relationships among previously disparate approaches.
2. It analyzes two different composition/reconstruction losses (L_0-1 vs. L_comp) and  connects them to the image formation model.
3. Broad ablation over key axes: sprite generator (pixels vs. MLP/U-Net), curriculum strategies, shared vs. sprite-specific transformations, decision module (Min-Loss vs. learned predictors), and regularizers.
4. Evaluations span eight clustering datasets of varied difficulty (digits, faces, traffic signs).

**Requested Changes:**

1. How sensitive is performance to the regularization weights (λ_freq, λ_bin, λ_empty) and Gumbel temperature?
2. Can you elaborate on the occlusion ordering module in your multi-layer variant
3. What is the speedup of the proposed predicted selection versus Min-Loss/greedy selection for typical K and L on CLEVR?

---

> ### Author Response · Authors · 2026-03-13
> **Addressing reviews and requested changes required by Reviewer 8ARL**
>
> We thank the reviewer for their constructive feedback and for identifying areas where our manuscript required further clarification. The following requested changes (RC) have been incorporated into the revised PDF (highlighted in blue).
>
> **[RC.1]** We thank the reviewer for their questions regarding the sensitivity of our model. We included a grid search analysis _in Appendix .2.2_ for the hyper-parameters $\lambda\_\text{freq}$​, $\lambda\_\text{bin}$ for clustering​​. We didn’t account for tuning Gumbel temperature as a hyper-parameter and used a learnable $\tau$.
>
> **[RC.2]** Our multi-layer variant adopts the occlusion formulation from DTI-Sprites (Monnier et al., 2021). The model predicts a binary occlusion matrix $\delta \in$ {0,1} $\^{L \times L}$, where $\delta\_{jl​}=1$ if layer $j$ occludes layer $l$. This formulation allows the model to explicitly reason about depth by predicting $\delta$, which effectively reorders the sprites. This ensures that each layer's appearance $o\_l^c$​ is correctly masked by its own transparency $o\_l^\alpha​$ and the transparency of all layers situated in front of it. We highlight the corresponding sentence *in Section 3.1.2*.
>
> **[RC.3]** We thank the reviewer for pointing out the need for more context in our performance analysis. Although not explicitly stated, the speedup comparison in Figure 7 is conducted on hypothetical experiments on the CLEVR dataset with increasing $K$ and $L$, where the  standard values are $K=6$, and $L=10$. We updated the text _in Section 5.2.2_ with the relevant information.

---

> > ### Comment · Reviewer_8ARL · 2026-04-28
> > **Regarding the experiments**
> >
> > I am satisifed with the response

---

### Review · Reviewer_4xmg · 2026-03-08

**Summary Of Contributions:**

This paper introduces a unified framework for sprite-based image models, breaking them down into four core modules: sprite generation, transformation, decision, and training criteria. Using this taxonomy, the authors conduct a systematic ablation study on clustering benchmarks to evaluate different design choices. Based on these findings, they propose a new prediction-based sprite model that scales linearly with the number of objects, bypassing the exponential search complexity of Min-Loss methods. The approach is then extended to multi-layer image decomposition and evaluated on synthetic datasets like Tetrominoes, Multi-dSprites, and CLEVR.

**Audience:**

Yes

**Audience Explanation:**

The paper will be highly relevant to researchers working on object-centric learning, unsupervised vision, and interpretable representations. This paper provides a highly readable, systematic breakdown of how explicit sprite-based models can be structured.

**Broader Impact Concerns:**

There are no major ethical concerns with this methodology paper.

**Claims And Evidence:**

No

**Claims Explanation:**

While a substantial portion of the paper is carefully executed, particularly the complexity analysis and the clustering ablations, the evidence does not fully support the overarching claims.

Most critically, the abstract states that the method performs on par with state-of-the-art unsupervised image segmentation methods on the standard CLEVR benchmark. However, Table 9 clearly shows that AST-Seg-B3-CT completely dominates on standard instance metrics, achieving 90.3 percent mIoU and 95.9 percent ARI-FG compared to the proposed method's 53.8 percent and 70.6 percent. The proposed method is only on par with DTI-Sprites specifically on class-aware metrics, which is a much narrower achievement.

Additionally, the mathematical formulation of the frequency regularizer in Equation 5 contradicts its stated purpose. The text states it penalizes using a sprite with a frequency lower than epsilon. However, the equation defines the penalty using a minimum function between the frequency and epsilon. Minimizing this term would force the frequency toward zero, actively encouraging unused sprites.

Finally, the authors state that regularization weights were selected through a grid search for each dataset to optimize performance. In a strictly unsupervised setting, using ground-truth labels to tune hyperparameters invalidates the unsupervised claim, and the protocol here is not explained clearly enough to verify fairness.

**Requested Changes:**

1. Recalibrate the CLEVR claims.

2. Correct Equation 5: The current formulation encourages zero frequency. If the intention was to penalize low frequency, this should be corrected. Please correct the equation and explicitly confirm whether the experiments were run using the flawed formulation or the intended one.

3. Clarify the hyperparameter tuning protocol. Explicitly describe how the grid search for regularization weights was conducted. If ground-truth labels were used to optimize performance in this unsupervised setting, this should be clearly justified and listed as a limitation.

4. Fix reporting inconsistencies in Table 8: Several entries, like DTI-Sprites on Multi-dSprites and CLEVR6, are reported without standard errors, despite the caption explicitly stating the table reports the mean and standard error over 3 runs.

---

> ### Author Response · Authors · 2026-03-13
> **Addressing reviews and requested changes required by Reviewer 4xmg**
>
> We thank the reviewer for their constructive feedback and for identifying areas where our manuscript required further clarification. The following requested changes (RC) have been incorporated into the revised PDF (highlighted in blue).
>
> **[RC.1]** We thank the reviewer for this suggestion. We revised our claims regarding CLEVR performance to more accurately reflect the results. Specifically, we will limit our claims of “being on-par with state-of-the-art on unsupervised class-aware image segmentation methods” (_in Abstract_, corresponding to mAcc and avg-mIoU). Note however that AST-Seg-B3-CT uses a very different methodology than all other methods to learn a background model, which is frozen during the other steps of training. We believe this is the main reason why it outperforms all other competitors by such a large margin, and that direct comparison to end-to-end image decomposition methods can be debated.
>
> **[RC.2]** We appreciate the reviewer pointing out the notation error in Equation 5. The formulation erroneously suggested a penalty for high frequency rather than a reward. We corrected _Equation 5_ to the following formulation:
>
>  $\mathcal{L}\_{\text{freq}} = \sum\_{k=1}^{K} \max \left( 0, \epsilon - \frac{1}{|D|} \sum\_{I} \sum\_{l=1}^{L} p\_{k,l}^I \right)$
>
> **Confirmation of Experiments:** We confirm that the experiments were conducted using the **intended** logic. Our implementation calculated the average frequency ($\mathcal{L}\_{\text{capped-freq}}$), capped it at $\epsilon$, and minimized the negative sum (1−$\mathcal{L}\_{\text{capped-freq}}$). This provided the correct gradient to push under-utilized clusters toward the threshold $\epsilon$. The error was strictly limited to the notation in the manuscript.
>
> **[RC.3]** We added a description of our grid search strategy and included an analysis of the performance of our model in clustering setup _in Appendix .2.2_. The weights $\lambda\_\text{freq}$​, and $\lambda\_\text{bin}$​​ for  clustering were respectively searched over the range \{0, 0.01, 0.1, 1.0\} and \{0, 0.001, 0.01, 0.1\}. For the multi-layer setup, we conduct a sequential grid search in range \{0, 0.0001, 0.001, 0.01, 0.1\} considering in order $\lambda\_\text{empty}$, $\lambda\_\text{freq}$​, and $\lambda\_\text{bin}$​​​. To establish a performance upper bound, we used ground truth labels to select the final hyperparameters. We acknowledge this as a limitation for fully unsupervised deployment and added a statement in corresponding Appendix section as a limitation.
>
> **[RC.4]** We apologize for the oversight regarding the reporting of standard errors. In the original manuscript, certain entries for DTI-Sprites were reported as single runs because preliminary results indicated that these configurations were not competitive, and further replication was not performed to avoid the computational and environmental cost of such experiments we deemed of low interest. To address the reviewer’s concern regarding clarity, we have updated _the caption of Table 8_ to explicitly state that grayed entries represent single-run results for sub-optimal configurations where initial performance did not warrant further replication. We believe this provides the necessary transparency while maintaining the focus on the competitive setup.

---

### Review · Reviewer_huXf · 2026-03-15

**Summary Of Contributions:**

This paper proposes a framework that encapsulates existing sprite-based image decomposition methods, breaking them down into four modules: sprite generation, transformation, decision, and reconstruction criteria. They do an in-depth exploration of this design space on small-scale tasks, and propose a novel and particularly scalable method. They replace the exponential-time "min-loss" search of prior methods with a gumbel-softmax decision module, which lets the method scale (more efficiently) to datasets like CLEVR.

**Audience:**

Yes

**Audience Explanation:**

The experimental methodology seems quite good, and their improvement upon the prior exponential search paradigm seems significant and worth highlighting.

**Claims And Evidence:**

Yes

**Claims Explanation:**

- Principled stepwise exploration, e.g., providing a principled reason for using an MLP for sprite generation
- Convincing qualitative evidence; the comparison between sprite-specific and shared transformations are convincing and support the architectural choices
- They effectively use small-scale experiments to inform the direction of their main experimental section, showing their choices generalize to larger/more complex tasks

**Requested Changes:**

- Kind of buries the lede; the final results seem significant, but seems somewhat buried compared to the (IMO) smaller contribution of creating a 4-module framework. Maybe some signposting to emphasize that the first experiments are just stepping stones to the more significant results
- Severe formatting issues/broken figures around page 16 need to be fixed
- Unclear what the takeaways from Table 2 should be, and the caption is highly redundant with the text. Why is pixel generation much better for MNIST?
- Terminology seems somewhat under-explained/dense to me -- ultimately just a formatting issue, but core concepts like one-by-one curriculum, linear mapping, and weight prediction could be highlighted more clearly / generally best to avoid requiring parsing dense equations to understand the mechanics

---

> ### Author Response · Authors · 2026-03-25
> **Addressing reviews and requested changes required by Reviewer huXf**
>
> We thank the reviewer for their constructive feedback and for identifying areas where our manuscript required further clarification. The following requested changes (RC) have been incorporated into the revised PDF (highlighted in blue).
>
> **[RC.1]** We thank the reviewer for this comment. We added a sentence in the *Introduction* to better emphasize the transition from core analysis on clustering to multi-layer application.
>
> **[RC.2]** We apologize for the broken figure in our revised PDF which we fixed in the second revised version.
>
> **[RC.3]** We thank the reviewer for this comment. We removed the redundancy in the caption and extended the comments in *Section 4.2*. We kindly do not agree with the reviewer’s assessment that the performance is much better on MNIST for the pixel-based approach compared to the latent-based approach. Indeed the results are very close (97.2 vs. 97.1) on MNIST and within the error margin on Colored MNIST. We simply believe both approaches perform similarly on MNIST because of its simplicity. Moreover the MLP-based approach is within the error margin of the best approach on all datasets and actually improve significantly the results on AffNIST and SVHN.
>
> **[RC.4]** We thank the reviewer for this comment. To improve clarity and make it easier for the reader to refer to the definitions, we have converted the first definitions from italic to bold-italic in the caption of *Figure 3*, *Section 3.3.1*, and *3.4*. We also added a sentence in the introduction of *Section 4*, directing the reader to *Figure 3* for a high-level overview of our terminology.

---

### Decision · Action_Editor_z3P4 · 2026-06-01

**Recommendation:** Accept with minor revision

**Additional Comments:**

After the authors' revision, most of the reviewers' concerns have been addressed. The reviewers unanimously recommend acceptance of the paper.

I find the work valuable and recommend its acceptance and publication.

Following the reviewers’ suggestions, as they prepare the final version, I encourage the authors to further emphasize the practical lessons learned from the large design-space study, as these insights are likely to have lasting value beyond the specific architecture proposed in the paper. It would also be beneficial to expand the discussion of the limitations associated with label-based hyperparameter selection and to consider evaluation protocols that more closely reflect fully unsupervised deployment settings. Finally, the authors are encouraged to carefully proofread the manuscript and address minor writing and formatting issues; for example, there is an extra space before “where” following Equation 5.

**Audience:**

Yes

**Audience Explanation:**

This paper may be of interest to researchers and practitioners working on object-centric learning, unsupervised representation learning, image clustering, image decomposition, and interpretable computer vision methods.

**Claims And Evidence:**

Yes

**Claims Explanation:**

Summary:

This paper presents a systematic study of sprite-based methods for unsupervised clustering and image decomposition. The authors introduce a unified framework that decomposes existing sprite-based approaches into four core components: sprite generation, transformation, decision, and training criteria. Using this taxonomy, they conduct an extensive empirical analysis of design choices across a diverse set of clustering benchmarks and identify key factors that influence performance. Building on these insights, the paper proposes a new prediction-based sprite selection mechanism that replaces the exponential-cost reassignment procedure used in prior Min-Loss approaches with a learned decision module based on Gumbel-Softmax. The resulting method scales linearly with the number of objects per image while maintaining competitive performance. Finally, the approach is extended to multi-layer image decomposition and evaluated on synthetic multi-object benchmarks. Overall, the paper contributes both a valuable empirical characterization of the sprite-model design space and a scalable sprite-based architecture that broadens the applicability of this family of interpretable object-centric methods.


Claims:

The paper makes several key claims: (1) sprite-based image models can be understood through a unified decomposition into four modular components, enabling systematic analysis of prior approaches; (2) many important architectural and training design choices can be effectively evaluated through clustering benchmarks, providing insights that transfer to more complex image decomposition settings; (3) the primary challenge in sprite-based methods lies in jointly learning and selecting sprites, and K-means-style selection mechanisms provide strong regularization and improved object discovery; and (4) a prediction-based sprite selection mechanism can achieve competitive performance while reducing the computational complexity of sprite selection from exponential to linear in the number of objects, making sprite-based approaches more scalable to realistic multi-object scenes.


Evidence:

The empirical evidence largely supports the paper’s claims. During the review process, several concerns were raised regarding the framing of the CLEVR results, a notation error in the frequency regularizer, hyperparameter tuning methodology, and reporting details. The authors provided thorough responses and revised the manuscript accordingly. In particular, they corrected the formulation of the regularizer, clarified that experiments were conducted using the intended implementation, revised claims regarding CLEVR performance to better match the reported results, and transparently documented the hyperparameter selection procedure, including the use of label-based model selection to establish an upper bound. These revisions significantly improved the clarity and accuracy of the paper, and the reviewers agreed that the revised claims are supported by the presented evidence.